# One Coin Has Two Sides: Single Positive Multi Label Learning from Salient Annotations

Xiaoyu Wang [1]   Zhuoming Li [2]   Bo Han [2]   Hui Liu [3]   Junhui Hou [4]   Yuheng Jia [2 3 5]

## Abstract

Single-Positive Multi-Label Learning (SPML) studies learning from incomplete supervision, where each instance is annotated with only one positive label despite potentially belonging to multiple categories. While existing methods assume the annotated labels are randomly distributed, real-world annotations are often biased toward the most salient category. We formalize this realistic scenario as Salient Single-Positive Multi-Label Learning (SalSPML). This salient annotation bias poses a challenge to conventional SPML methods, as the missing labels often correspond to less salient and harder-to-recognize categories. Fortunately, we find that salient annotations are typically more representative and informative. Motivated by this insight, we propose Prototype-Guided Rejection for Salient Annotation (PiSA), which constructs reliable classwise prototypes from salient labels and leverages them to guide embedding learning for recognition of non-salient labels. We theoretically demonstrate that SalSPML is harder than Random SPML due to irreducible annotation bias, and under SalSPML, more accurate prototypes facilitate false-negative label detection. Experiments on multiple benchmarks, together with two newly constructed real-world SalSPML datasets, demonstrate that PiSA consistently outperforms existing methods, achieving an average mAP improvement of 3.16%.

[1]College of Software Engineering, Southeast University, Nanjing, China [2]School of Computer Science and Engineering, Southeast University, Nanjing, China [3]School of Computing and Information Sciences, Saint Francis University, Hong Kong, China [4]Department of Computer Science, City University of Hong Kong, Hong Kong, China [5]Key Laboratory of New Generation Artificial Intelligence Technology and Its Interdisciplinary Applications (Southeast University), Ministry of Education, China. Correspondence to: Yuheng Jia <yhjia@seu.edu.cn>.

*Proceedings of the 43rd International Conference on Machine Learning*, Seoul, South Korea. PMLR 306, 2026. Copyright 2026 by the author(s).

## 1. Introduction

Multi-Label Learning (MLL) aims to assign multiple labels to each instance and has been widely applied in various real-world scenarios such as medical diagnosis (Liu et al., 2022), emotion recognition (Huang et al., 2025) and recommender systems (Thang et al., 2025). However, obtaining complete and accurate label sets for all instances is labor-intensive and costly. To alleviate this annotation bottleneck, various weakly supervised multi-label learning paradigms have been extensively explored (Han et al., 2026; Li et al., 2025b). Among these, Single Positive Multi-Label Learning (SPML), where each instance is annotated with only one of its relevant labels, has attracted increasing attention as a practical form of weak supervision (Cole et al., 2021).

Existing SPML methods have demonstrated promising performance by leveraging techniques such as loss correction, implicit label inference, and confidence-aware learning (Cole et al., 2021; Zhou et al., 2022; Kim et al., 2023; Chen et al., 2024a; Gharib et al., 2025). A common underlying assumption in these approaches is that the observed positive label is generated through a random selection process from the true label set. Under this assumption, positive-label omissions are assumed to occur at random, enabling unbiased risk estimation.

However, this assumption rarely holds in practice. Real-world annotators do not behave randomly; instead, they tend to annotate only the most salient or visually prominent objects while overlooking subtle but semantically relevant ones. We formalize this more realistic setting as *Salient Single Positive Multi-label Learning* (SalSPML), where only the most noticeable positive label is retained. When applied to SalSPML, existing SPML methods suffer substantial performance degradation. As shown in Figure 1, under the SalSPML, these methods show significantly worse performance. We theoretically show that salient annotation introduces a systematic *annotative bias* that does not vanish with more data and directly inflates the excess risk bound (see Section 3.2).

Despite its challenges, SalSPML also reveals a valuable opportunity. As shown in Figure 2, although annotations are limited in number, salient labels are typically more rep-

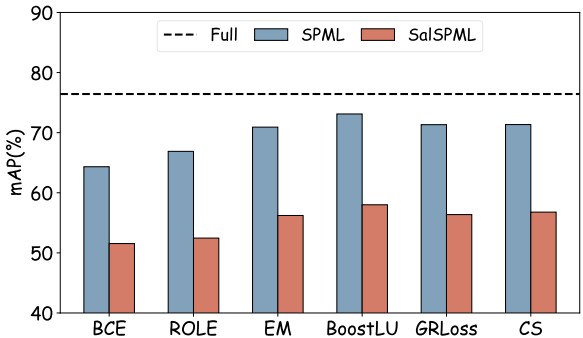

*Figure 1.* Performance comparison on the COCO dataset under two different settings. Binary Cross-Entropy (BCE) is a general method for multi-label learning, ROLE (Cole et al., 2021), EM (Zhou et al., 2022), BoostLU (Kim et al., 2023), GRLoss (Chen et al., 2024b) and CS (Gharib et al., 2025) are representative for single-positive multi-label learning methods. All of these methods experience significant performance drops in SalSPML.

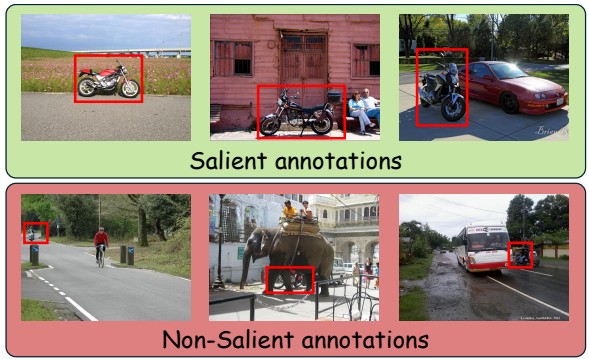

*Figure 2.* Observation of salient and non-salient annotations for the motorbike class in our real-world dataset. Non-salient instances are frequently missed due to visual degradation factors such as occlusion, small relative size, or complex backgrounds.

resentative and reliable, often corresponding to prototypical instances of their categories. This observation suggests that salient labels, while biased, may provide high-quality semantic anchors.

Motivated by this insight, we construct feature prototypes from salient annotations to guide representation learning and simultaneously recover labels for non-salient instances. Specifically, we propose a novel method called **P**rototype **G**u**i**ded Rejection for **S**alient **A**nnotation (**PiSA**), which supervises class embedding learning by aligning salient label embeddings with their corresponding prototypes while pushing them away from other prototypes. This alignment enhances class discrimination. To address annotation bias, PiSA exploits the semantic consistency between sample embeddings and class prototypes, leading to more reliable non-salient label detection compared to confidence-based approaches (see Section 4.5). Based on this principle, we further introduce a prototype-driven rejection mechanism

to suppress uncertain or noisy signals. We theoretically show that more accurate class prototypes lead to improved recovery of non-salient labels (see Section 3.5).

Our main contributions are summarized as follows:

- **SalSPML Setting.** To the best of our knowledge, we are the first to introduce Salient Single Positive Multi-Label Learning (SalSPML), a realistic annotation setting. Our theoretical analysis shows that, unlike existing SPML settings, SalSPML introduces an additional sampling-induced annotation bias, which can lead to systematic performance degradation, as further verified empirically.

- **PiSA Method.** We propose **PiSA**, which constructs class prototypes from salient annotations to recover non-salient labels. We theoretically prove that prototype guidance mitigates the annotation bias introduced by SalSPML, and that higher-quality prototypes yield greater bias reduction. Experiments on multiple benchmarks consistently demonstrate state-of-the-art performance with clear margins.

- **Real-World SalSPML Datasets.** We construct two real-world datasets that instantiate the SalSPML setting, reflecting salient annotation patterns in practice.

## 2. Related Work

### 2.1. Multi-Label Learning with Partial Labels

Multi-label learning has achieved notable success (Liu et al., 2021), but most methods rely on fully annotated datasets, which are expensive and impractical in real-world applications. In practice, annotators often miss relevant categories, especially when the label space is large, leading to incomplete annotations (Sun et al., 2010).

To address this, Multi-Label Learning with Partial Labels (ML-PL) has attracted increasing attention, in which only partial positive labels are observed (Bucak et al., 2011; Sun et al., 2010). Early work (Durand et al., 2019) proposed normalized BCE loss with curriculum learning to handle missing labels, though its sensitivity to noise was later highlighted (Ma et al., 2020; Wang et al., 2021; Zhou et al., 2021). Asymmetric Loss (Ridnik et al., 2021) mitigates label imbalance and improves robustness. Recent advances include loss-aware strategies such as LL-R (Kim et al., 2022), which rejects high-loss samples, and semantic-guided methods like HST (Chen et al., 2024a) that explore semantic relationships among labels to transfer knowledge from known to unknown labels.

## 2.2. Single Positive Multi-Label Learning

Single Positive Multi-Label Learning (SPML) was first formalized by (Cole et al., 2021), where each training instance is labeled with only one positive label, and all other labels remain unobserved. Due to the lack of comprehensive supervision, this method estimates the expected number of true positives and applies reweighting strategies under the assumption that unobserved labels are negative (AN assumption). Other approaches challenge the AN assumption altogether. For example, EM (Zhou et al., 2022) treats all unannotated labels as unknown and employs an entropy-maximizing objective along with a heuristic asymmetric pseudo-labeling strategy. OPML (Li et al., 2025a) addresses performance degradation in SPML by enforcing pairwise label separation and applying high-rankness constraints to reduce negative label dominance. CLS (Lyu et al., 2025) identifies missing annotations from the perspective of sample selection by simultaneously learning two models to supervise each other.

In summary, although these methods achieve competitive results in ML-PL settings, they often rely heavily on the model's own prediction confidence for supervision, making them vulnerable in SalSPML, where missing labels are particularly difficult to distinguish.

## 3. Methodology

In this section, we first analyze the SalSPML setting and show that training with salient supervision introduces an irreducible annotation bias (Subsection 3.2). Despite this limitation, salient labels also provide a unique advantage: they typically correspond to prominent and representative objects, enabling the learning of high-quality class prototypes. We build upon this observation and propose a class-wise embedding encoder together with a prototype-guided learning scheme that leverages these prototypes to guide representation learning and recover suppressed non-salient signals. Furthermore, we provide theoretical analysis demonstrating that higher-quality prototypes lead to greater bias reduction (Subsection 3.5). The overall framework is illustrated in Figure 3.

### 3.1. Preliminaries

Let $x \in \mathcal{X}$ denote an input instance (e.g., an image) and $Y \subseteq \{1, \ldots, M\}$ denote its (latent) full set of ground-truth positive labels, where $M$ denotes the number of categories. We consider a hypothesis class $\mathcal{F}$ of predictors $f : \mathcal{X} \to \mathbb{R}^M$ and a non-negative loss function $L(f(x), j)$ measuring the prediction error on class $j$.

**Definition 3.1** (Population Risk). To characterize the effect of missing labels, the population risk of a predictor $f$ is defined as

$$R(f) = \mathbb{E}_{(x,Y)\sim\mathcal{D}}\Big[\sum_{j\in Y} L\big(f(x), j\big)\Big], \quad (1)$$

where $\mathcal{D}$ denotes the true data distribution.

Given $(x, Y)$, we consider two annotation mechanisms that observe a single positive label:

**Random annotation.** Most existing SPML settings assume this uniform random selection of the observed positive label. A label $y^{\mathrm{rnd}}$ is sampled uniformly at random from the positive label set $Y$, i.e.,

$$\mathbb{P}(y^{\mathrm{rnd}} = j \mid x, Y) = \frac{1}{|Y|}, \quad \forall j \in Y. \quad (2)$$

**Salient annotation.** A label $y^{\mathrm{sal}}$ is selected according to a (possibly instance-dependent) saliency function

$$y^{\mathrm{sal}} = \phi(x, Y) \in Y, \quad (3)$$

which deterministically returns the most salient positive label. All remaining labels are unobserved.

### 3.2. Bias Induced by Salient Annotations

We show that SalSPML introduces a systematic bias that leads to an underestimation of Population Risk.

**Random versus Salient supervision.** Under Random SPML, the observed supervision $y^{\mathrm{rnd}}$ is sampled uniformly from the ground-truth positive label set $Y$. Conditioned on $(x, Y)$, the expected loss satisfies

$$\mathbb{E}\big[L(f(x), y^{\mathrm{rnd}}) \mid (x, Y)\big] = \frac{1}{|Y|}\sum_{j\in Y} L(f(x), j), \quad (4)$$

which in expectation equals the average positive risk conditioned on $(x, Y)$, and is therefore proportional to the summed positive risk up to the factor $1/|Y|$.

In contrast, SalSPML is governed by label saliency. Given $(x, Y)$, the annotation process reveals a single salient label $y^{\mathrm{sal}} \in Y$, while the remaining labels $Y^{\mathrm{hard}} = Y \setminus \{y^{\mathrm{sal}}\}$ remain unannotated. As a result, the loss contributions of non-salient labels are consistently omitted. The risk induced by salient supervision is therefore given by

$$\begin{aligned} R_{\mathrm{sal}}(f) &= \mathbb{E}\big[L(f(x), y^{\mathrm{sal}})\big] \\ &= R(f) - \Delta_{\mathrm{bias}}(f), \end{aligned} \quad (5)$$

where

$$\Delta_{\mathrm{bias}}(f) = \mathbb{E}_{(x,Y)\sim\mathcal{D}}\left[\sum_{j\in Y^{\mathrm{hard}}} L(f(x), j)\right] \quad (6)$$

explicitly quantifies the annotative bias arising from the consistently unobserved, non-salient positive labels.

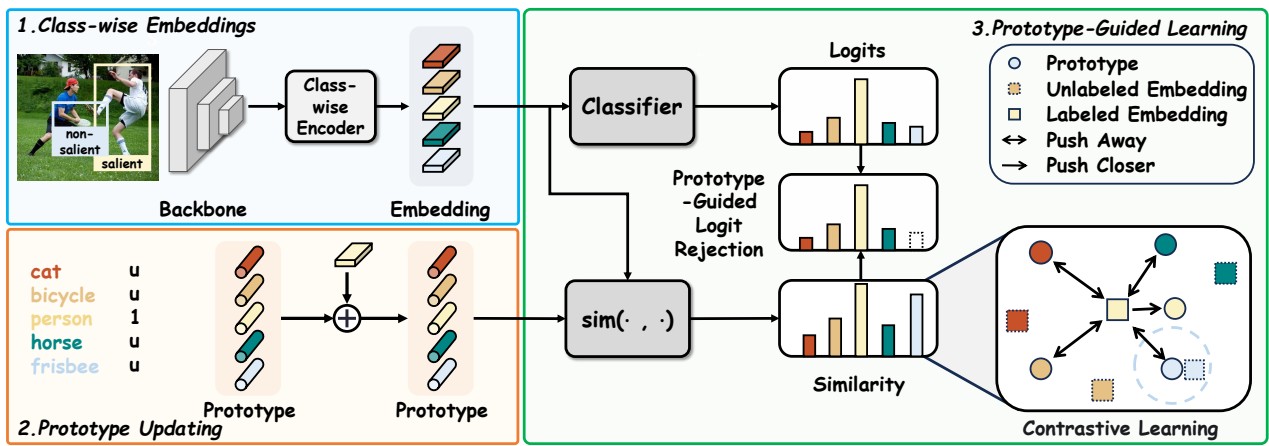

*Figure 3.* Overview of the proposed method. Given salient-labeled samples, the model generates class-wise embeddings and constructs reliable class prototypes from salient label's embeddings. Here, label=1 indicates the class is annotated as positive, while label=u indicates the class is not annotated (unlabeled). These prototypes are used to guide embedding learning and identify potential non-salient labels.

**Theorem 3.2** (Excess Risk with Salient Supervision). *Let $\hat{f} = \arg\min_{f \in \mathcal{F}} \hat{R}_{\text{sal}}(f)$ be the empirical risk minimizer trained on $n$ saliently annotated samples. With probability at least $1 - \delta$,*

$$R(\hat{f}) - R(f^\star) \leq \mathcal{O}(\mathfrak{R}_n(\mathcal{F})) + \mathcal{O}\left(\sqrt{\frac{\log(1/\delta)}{n}}\right) + \Delta_{\text{bias}}(\hat{f}), \quad (7)$$

*where $f^\star$ denotes the minimizer of the true population risk.*

As the sample size $n$ increases, the estimation terms vanish, while the bias term $\Delta_{\text{bias}}(\hat{f})$ persists. This shows that standard Empirical Risk Minimization under salient supervision is inherently biased and insufficient for minimizing the true population risk.

### 3.3. Class-Wise Embedding Construction

Multi-label learning models usually consist of two modules. The first module is the backbone responsible for deriving a feature map $F \in \mathbb{R}^{H \times W \times C}$ for each sample, where $H$, $W$, and $C$ are height, width, and number of channels respectively. Afterwards, $F$ is processed by average global pooling to obtain a low-dimensional embedding $Z$. The second module contains a linear layer that converts this low-dimensional embedding into the final prediction.

In the SalSPML setting, non-salient objects tend to be weakly expressed in image representations. As a result, the standard practice of extracting a single global feature vector per image (Bai et al., 2021; Chen & Yeh, 2024) can overly bias the representation toward salient objects. To address this, we introduce class-wise embeddings. Following ML-Decoder (Ridnik et al., 2023), we first use a backbone to extract a feature map, and then apply a multi-head attention module to produce class-wise embeddings $E \in \mathbb{R}^{M \times D}$, where $M$ denotes the number of classes and $D$ is the embed-

ding dimension. A class-specific linear projection is applied to each embedding to output the predicted probability.

### 3.4. Prototype-Guided Embedding Learning

We exploit a key advantage of SalSPML: salient objects are typically prominent and easy to recognize, and their embeddings tend to capture the most representative and class-distinctive characteristics. As a result, embeddings associated with salient labels provide reliable semantic anchors for each class. This observation motivates us to leverage salient labels not only as supervision signals, but also as a means to establish stable class-level references. Specifically, by aggregating embeddings from salient samples, we can learn class prototypes that emphasize discriminative semantics while suppressing noisy or incidental features. These prototypes serve as a global representation of each class and can further assist in identifying non-salient objects.

Formally, we maintain a prototype $Q^c$ for each class $c$, initialized as a zero vector, which is updated using class-wise embeddings from salient labels:

$$Q^c \leftarrow \text{Normalize}\left(Q^c + E_i^c\right), \quad \text{if } \hat{y}_i^c = 1, \quad (8)$$

where $E_i^c$ denotes the class-wise embedding of instance $x_i$ for class $c$, and $\text{Normalize}(\cdot)$ applies $L_2$ normalization.

Once reliable class prototypes are obtained, they provide a natural supervision signal for representation learning. Intuitively, embeddings associated with salient labels should be close to the prototype of their own class, while remaining dissimilar to prototypes of other classes. To capture this principle, for each instance $i$, we measure both the similarity to the matched prototype and the similarity to mismatched

prototypes:

$$s_i^+ = \frac{1}{\sum_{c=1}^{M} \mathbb{I}\left[\hat{y}_i^c = 1\right]} \sum_{c=1}^{M} \mathbb{I}\left[\hat{y}_i^c = 1\right] \left\langle E_i^c, Q^c \right\rangle, \quad (9)$$

$$s_i^- = \frac{1}{\sum_{c=1}^{M} \mathbb{I}\left[\hat{y}_i^c = 1\right]} \sum_{c=1}^{M} \mathbb{I}\left[\hat{y}_i^c = 1\right] \frac{\sum_{k \neq c} \left\langle E_i^c, Q^k \right\rangle}{M-1}, \tag{10}$$

where $s_i^+$ in Eq. (9) represents the average similarity between the class-wise embedding in the positive label set and the corresponding class prototype, $s_i^-$ in Eq. (10) denotes the average similarity between this salient class-wise embedding and all other class prototypes.

Based on these similarity scores, we define a prototype contrastive loss:

$$\mathcal{L}_{\text{PL}} = \frac{1}{N} \sum_{i=1}^{N} \left( \left(1 - s_i^+\right) + \gamma s_i^- \right), \tag{11}$$

which explicitly encourages salient embeddings to align with their matched prototypes while remaining separated from mismatched ones. By minimizing Eq. (11), class prototypes act as semantic anchors that encourage salient embeddings to align with their matched prototypes while remaining separated from mismatched ones. This supervision strengthens class-discriminative representations and enables knowledge transfer to non-salient samples via prototype-guided alignment.

### 3.5. Mitigating Bias via Prototype Guidance

Theorem 3.2 shows that, under salient supervision, ERM incurs an irreducible excess risk term $\Delta_{\text{bias}}(\hat{f})$. Therefore, our goal is to attenuate this annotation bias. To this end, we use prototype matching to recover non-salient labels with probability $\eta \in [0, 1]$, thereby reducing the effective bias term.

**Theorem 3.3** (Bias Reduction via Prototype Recovery)**.** *Let $\mu_c$ denote the true mean of the class in the embedding space for the class $c$, and let $P_c$ be the corresponding learned prototype satisfying $\|P_c - \mu_c\| \leq \epsilon$. Assume a margin condition $\gamma > 0$ such that true positive embeddings are, in expectation, more similar to their matched prototypes than to mismatched ones by at least $\gamma$. Then prototype matching recovers each hard positive with probability at least $\eta \geq 1 - \exp\left(-c(\gamma - \epsilon)^2\right)$ for some constant $c > 0$, and the bias term is reduced to $(1 - \eta)\Delta_{\text{bias}}(f)$.*

This result shows that the bias will be reduced as the prototype accuracy improves: a smaller approximation error $\epsilon$ leads to a higher recovery rate $\eta$, thus further reducing the residual annotation bias.

### 3.6. Prototype-Guided Loss Rejection

Non-salient positives are easily missed under SalSPML and may be treated as negatives during training, introducing false negative supervision. We address this issue with *Prototype-Guided Loss Rejection* (PGLR), which uses class prototypes as semantic references to identify such false negatives. Specifically, we measure the similarity between each class-wise embedding and its corresponding prototype:

$$G_i^j = \left\langle E_i^j, Q^j \right\rangle, \tag{12}$$

where $G_i^j$ denotes the similarity between the class-wise embedding $E_i^j$ of sample $x_i$ and prototype $Q^j$.

To identify potential false negative labels and exclude them from training, we design a criterion based on the similarity matrix $G$. Specifically, for each sample, unlabeled classes with top-$k$ prototype similarities are regarded as potential false negatives and their negative losses are rejected. We define the negative-loss mask as

$$R_i^j = \mathbb{I}\left[G_i^j < \text{top}_k(G_i)\right], \tag{13}$$

where $R_i^j = 0$ indicates that the negative loss of class $j$ is rejected.

The corresponding loss function is formulated as follows:

$$\mathcal{L}_{\text{PGLR}} = -\frac{1}{BM} \sum_{i=1}^{B} \sum_{j=1}^{M} \left[ \mathbb{I}\left[\hat{y}_i^j = 1\right] \cdot \log\left(p_i^j\right) \right. \\ \left. + \mathbb{I}\left[\hat{y}_i^j = u\right] \cdot R_i^j \cdot \log\left(1 - p_i^j\right) \right]. \tag{14}$$

This loss excludes unobserved classes with high similarity to class prototypes from optimization, thereby reducing the impact of false negative labels.

### 3.7. Overall Objective

Since prototype-based supervision is meaningful only after prototypes become reliable, we employ a warm-up stage with the standard binary cross-entropy loss:

$$\mathcal{L}_{\text{BCE}} = -\frac{1}{BM} \sum_{i=1}^{B} \sum_{j=1}^{M} \left[ \mathbb{I}\left[\hat{y}_i^j = 1\right] \cdot \log\left(p_i^j\right) \right. \\ \left. + \mathbb{I}\left[\hat{y}_i^j = u\right] \cdot \log\left(1 - p_i^j\right) \right]. \tag{15}$$

During warm-up, the model is optimized by Eq. (15) while updating feature prototypes. After $T_w$ warm-up epochs (we set $T_w = 1$ to avoid early overfitting), we move to the second stage, where we incorporate both the prototype learning loss and the Prototype-Guided Loss Rejection. Specifically, the objective in the second stage is:

$$\mathcal{L}_{\text{PiSA}} = \mathcal{L}_{\text{PGLR}} + \alpha \mathcal{L}_{\text{PL}}. \tag{16}$$

The detailed training algorithm is provided in Algorithm 1.

**Algorithm 1** Algorithm of PiSA

**Input**:
$\mathcal{D}$: multi-label dataset with salient annotations;
$T_{\mathrm{w}}$: warm up epoch;
$T_{\max}$: the total training epoch;
$k, \alpha, \gamma$: the parameters of the loss function.
**Output**:
network $f(\cdot \mid \theta)$.
1: **for** $t = 1$ to $T_{\max}$ **do**
2:     Sample a mini-batch from $\mathcal{D}$;
3:     **if** $t \leq T_{\mathrm{w}}$ **then**
4:         **Stage 1:**
5:         Calculate the loss according to Eq. (15);
6:     **else**
7:         **Stage 2:**
8:         Calculate the similarity $G$ according to Eq. (12);
9:         Calculate $R$ according to Eq. (13);
10:       Calculate the loss according to Eq. (16);
11:     **end if**
12:     Update the class prototype according to Eq. (8);
13:     Update the model parameters;
14: **end for**

# 4. Experiments

## 4.1. Data Preparation

**Real-world salient annotation.** To reflect realistic annotation behavior, we build two SalSPML datasets from MS-COCO (Lin et al., 2014) and VOC (Everingham et al., 2010) via manual salient labeling. Specifically, we recruited 10 annotators, who selected the single most visually salient object for each image as the supervision label. This process results in a real-world benchmark that closely matches the SalSPML setting.

**Synthetic salient annotation.** In addition to real-world data, we construct synthetic SalSPML datasets based on MS-COCO, VOC, and NUS-WIDE (Chua et al., 2009). We adopt two label selection strategies:

- **confidence-based**: We train a classifier on fully annotated data and retain, for each image, the positive label with the highest prediction confidence.

- **size-based**: For datasets with instance or semantic segmentation annotations (COCO and VOC), we select the label corresponding to the largest region in the image as the single positive label.

Additionally, we use a pretrained CLIP model (Radford et al., 2021) to construct SalSPML datasets by selecting, for each image, the class with the highest CLIP confidence as the salient annotation; results are reported in Appendix E.

## 4.2. Experimental Setting

We use Adam with a learning rate of $1 \times 10^{-5}$ and a batch size of 16 for all datasets. Unless otherwise specified, we fix all hyperparameters across experiments: for our method, we set the loss balancing coefficients $\alpha = \gamma = 1$ and the label rejection parameter $k = 3$. We train for a total of $T_{\max} = 20$ epochs with a minimal warm-up of $T_w = 1$. This early intervention is adopted because models tend to rapidly overfit to salient concepts, making immediate bias mitigation more beneficial. All results are averaged over three runs with different random seeds and reported as mean mAP $\pm$ standard deviation. We further conduct paired $t$-tests between PiSA and each baseline at a significance level of 0.05 over the multiple runs, and report the corresponding Win/Tie/Loss counts.

For all datasets, 20% of the training data is held out for validation. Both the validation and test sets remain fully labeled. We select the model checkpoint that achieves the best performance on the validation set, and report its results on the test set as the final performance. All experiments are conducted using ResNet-50 (He et al., 2016) pretrained on ImageNet, implemented in PyTorch 1.13.1 and run on an NVIDIA GeForce RTX 4090 GPU.

## 4.3. Comparison Methods

In the SalSPML setting, we compare PiSA with 10 baseline methods, grouped into three categories: (1) standard multi-label classification methods, including **BCE**, **ASL** (Ridnik et al., 2021), and **ML-Decoder** (Ridnik et al., 2023); (2) partial-label multi-label learning methods, including **LL-R** (Kim et al., 2022) and **BoostLU** (Kim et al., 2023); and (3) single-positive multi-label learning methods, including **ROLE** (Cole et al., 2021), **EM** (Zhou et al., 2022), **GR-Loss** (Chen et al., 2024b), and **CS** (Gharib et al., 2025).

## 4.4. Comparison Results

**Performance in Real-World Benchmarks.** We conduct our first experiment on two manually annotated salient single-positive benchmarks. We compare PiSA with representative SPML baselines under the same evaluation metric.

As shown in Table 1, PiSA achieves the best performance on both COCO and VOC. In particular, PiSA improves over the strongest baseline by +4.05% on COCO (59.58% vs. 55.53%) and +1.48% on VOC (87.83% vs. 86.35%), while maintaining comparable variance. Moreover, under a paired t-test, PiSA achieves **wins in all 20 cases**. These results demonstrate the effectiveness and robustness of PiSA under realistic salient single-positive annotation settings.

**Performance on Synthetic Datasets.** We evaluate PiSA under two synthetic label construction strategies in Ta-

*Table 1.* Comparison with state-of-the-art methods on our real SalSPML datasets (mAP, higher is better). **Full** indicates a fully-labeled upper bound (BCE trained). **Avg.** denotes the average of the COCO and VOC results, and **Sig.** denotes win/tie/loss counts of PiSA against each baseline based on a paired t-test.

| Method | COCO | VOC | Avg. | Sig. |
|---|---|---|---|---|
| Full | 75.88±0.11 | 88.98±0.08 | 82.43 | 2 / 0 / 0 |
| BCE | 47.82±0.19 | 82.46±0.36 | 65.14 | 2 / 0 / 0 |
| ASL | 50.36±0.79 | 84.60±0.63 | 67.48 | 2 / 0 / 0 |
| ML-Decoder | 56.11±0.29 | 83.77±0.75 | 69.94 | 2 / 0 / 0 |
| ROLE | 47.09±0.25 | 85.51±0.57 | 66.30 | 2 / 0 / 0 |
| EM | 50.66±0.21 | 86.04±0.32 | 68.35 | 2 / 0 / 0 |
| LL-Ct | 54.88±0.78 | 82.78±0.38 | 68.83 | 2 / 0 / 0 |
| LL-R | 55.28±0.17 | 85.22±0.41 | 70.25 | 2 / 0 / 0 |
| BoostLU | 54.24±0.27 | 84.77±0.14 | 69.50 | 2 / 0 / 0 |
| GRLoss | 51.53±0.34 | 86.35±0.09 | 68.94 | 2 / 0 / 0 |
| CS | 55.33±0.43 | 85.17±0.48 | 70.25 | 2 / 0 / 0 |
| PiSA | **59.58±0.29** | **87.83±0.45** | **73.71** | 20 / 0 / 0 |

ble 2. Overall, PiSA achieves the best performance across all datasets and both construction strategies. Under the confidence-based setting, PiSA improves over the second-best method by +1.37% on VOC, +3.37% on COCO, and +0.46% on NUS-WIDE. Under the size-based setting, PiSA further yields gains of +4.10% on VOC and +3.02% on COCO. Averaged over all settings, PiSA brings a +3.16% absolute improvement over the best competitor. Notably, according to the significance test, PiSA achieves **48** wins and only **2** ties out of **50** cases, indicating consistent and statistically supported improvements over the baselines.

**Performance under Partial Labels.** To evaluate effectiveness under varying supervision levels, we conduct experiments on Visual Genome (Krishna et al., 2017) using a global label-retention setting. As shown in Tab. 3, our method consistently outperforms baselines across all retention rates. Specifically, it improves mAP by +0.40%, +0.83%, and +1.07% at 30%, 50% and 70% retention, respectively, achieving the largest gain (+2.32% relative) at 70%. The significance analysis further confirms our superiority in **all 21 cases**.

### 4.5. Further Analyses

**Ablation study.** To demonstrate the effectiveness of all components, we conduct an ablation study using the VOC, COCO, and NUS-WIDE datasets under the confidence-based setting. As shown in Table 4, both prototype-guided embedding learning (PL) and prototype-guided loss rejection (PGLR) effectively enhance the model performance. To be specific, the prototype-guided embedding loss improves the mAP by an average of 1.48% across the three datasets, while the prototype-guided loss rejection brings an average gain of 1.12%. When combined, these two components lead to a 2.93% improvement, indicating that incorporating both

terms yields the best overall performance.

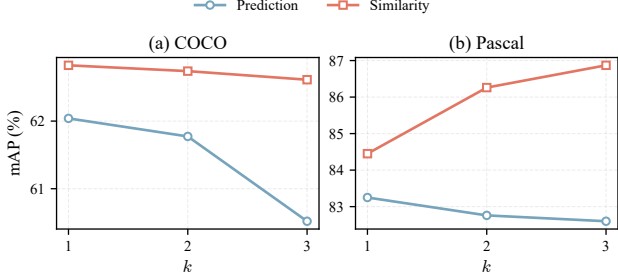

*Figure 4.* Comparison of two label rejection strategies for mitigating false negative supervision. We report mAP under the same setting.

**Our rejection method is better.** We compare two label rejection strategies with all other settings fixed: rejecting the top-$k$ unlabeled categories based on (1) prediction probability or (2) embedding–prototype similarity, where $k \in \{1, 2, 3\}$. As shown in Figure 4, the similarity-based strategy consistently performs better and is more stable across datasets and $k$.

Compared with prediction-based rejection, similarity-based rejection yields consistent mAP improvements on both datasets. On VOC, it achieves gains of +1.20%, +3.50%, and +4.27% for $k=1/2/3$, respectively. On COCO, the corresponding gains are +0.79%, +0.97%, and +2.09%. Notably, the probability-based strategy degrades as $k$ increases (especially at $k=3$ on COCO), whereas the similarity-based strategy remains stable, indicating stronger robustness to the rejection strength.

We further evaluate an adaptive rejection rule in Table 5. For each image, it rejects an unlabeled class when its prototype similarity exceeds $\mu + w\sigma$, where $\mu$ and $\sigma$ are the mean and standard deviation of all class-prototype similarities for that image, and $w$ controls the threshold. The results are comparable to the fixed top-$k$ strategy, while $k = 3$ remains slightly better and simpler, indicating that PiSA is robust to the rejection rule.

**PiSA reduces annotation bias.** We directly measure the annotation bias term induced by salient supervision. Concretely, we instantiate the loss $L(\cdot, \cdot)$ in Eq. (6) with the BCE loss and compute the corresponding bias estimate on both our real SalSPML datasets (Real dataset) and confidence-based synthetic SalSPML datasets (Conf. dataset). Figure 5 reports the measured $\Delta_{\text{bias}}$ values (lower is better). Across all settings and datasets, PiSA consistently yields the smallest bias. For example, on the real COCO setting, PiSA reduces the measured bias from 5.60 (BCE) to 4.57, and on the real VOC setting from 3.40 to 2.91. A similar trend holds under confidence-based SalSPML: PiSA reduces the

*Table 2.* Results under two synthetic SalSPML construction strategies: confidence-based (highest-confidence positive) and size-based (largest-region positive).

| Method | confidence-based | | | size-based | | Avg. | Sig. |
| | VOC | COCO | NUS | VOC | COCO | | |
|---|---|---|---|---|---|---|---|
| Full | 88.98±0.08 | 75.88±0.11 | 50.18±0.29 | 88.98±0.08 | 75.88±0.11 | 75.98 | - |
| BCE | 80.75±0.38 | 51.55±0.32 | 31.59±0.52 | 72.70±0.19 | 45.31±0.20 | 57.86 | 5 / 0 / 0 |
| ASL (ICCV 2021) | 83.06±1.17 | 53.77±0.10 | 34.29±0.52 | 80.08±0.63 | 46.81±0.55 | 59.60 | 5 / 0 / 0 |
| ROLE (CVPR 2021) | 83.86±0.37 | 52.46±0.30 | 31.16±0.47 | 80.82±0.76 | 45.09±0.33 | 58.68 | 5 / 0 / 0 |
| LL-Ct (CVPR 2022) | 79.31±0.79 | 57.19±0.18 | 34.65±0.19 | 73.70±0.41 | 49.77±0.15 | 58.92 | 5 / 0 / 0 |
| LL-R (CVPR 2022) | 82.51±0.49 | 57.06±0.27 | 34.61±0.33 | 77.27±0.21 | 49.55±0.09 | 60.20 | 5 / 0 / 0 |
| EM (ICCV 2021) | 84.57±0.47 | 56.22±0.22 | 34.78±0.09 | 79.35±0.45 | 48.15±0.34 | 60.61 | 5 / 0 / 0 |
| ML-Decoder (WACV 2023) | 82.82±0.53 | 60.13±0.48 | 36.00±0.32 | 82.09±1.09 | 50.92±0.07 | 62.39 | 5 / 0 / 0 |
| BoostLU (CVPR 2023) | 81.74±1.16 | 58.00±0.42 | 34.84±0.73 | 72.77±0.81 | 50.28±0.61 | 59.53 | 5 / 0 / 0 |
| GRLoss (IJCAI 2024) | 85.02±0.63 | 56.37±0.33 | 35.77±0.37 | 81.41±0.08 | 48.70±0.27 | 61.45 | 4 / 1 / 0 |
| CS (TMLR 2025) | 83.45±0.40 | 56.78±0.57 | 37.02± 0.27 | 82.32±0.41 | 49.86±0.15 | 61.89 | 4 / 1 / 0 |
| Ours | **86.39±0.64** | **63.50±0.16** | **37.48±0.14** | **86.42±0.19** | **53.94±0.07** | **65.55** | 48 / 2 / 0 |

*Table 3.* Performance comparison of our model with other methods under different positive label retention rates on the VG dataset.

| Method | Retention Rate | | | Sig. |
| | 30% | 50% | 70% | |
|---|---|---|---|---|
| BCE | 38.40±0.09 | 40.90±0.07 | 42.25±0.15 | 3 / 0 / 0 |
| ASL | 39.25±0.03 | 42.00±0.05 | 43.54±0.03 | 3 / 0 / 0 |
| LL-R | 40.91±0.11 | 43.28±0.08 | 44.91±0.05 | 3 / 0 / 0 |
| LL-Ct | 40.67±0.08 | 43.26±0.07 | 44.84±0.05 | 3 / 0 / 0 |
| EM | 37.58±0.05 | 39.36±0.12 | 39.82±0.07 | 3 / 0 / 0 |
| ML-Decoder | 41.25±0.05 | 44.73±0.07 | 46.95±0.09 | 3 / 0 / 0 |
| BoostLU | 41.34±0.06 | 44.28±0.08 | 46.17±0.07 | 3 / 0 / 0 |
| Ours | **41.74±0.08** | **45.11±0.08** | **47.24±0.05** | 21 / 0 / 0 |

*Table 4.* Comparison of mAP (%) with and without the PL and PGLR modules on COCO, NUS-WIDE and VOC under the Sal-SPML setting.

| w/ PL | w/ PGLR | COCO | NUS | VOC | Avg. |
|---|---|---|---|---|---|
| | | 59.59 | 35.68 | 83.21 | 59.49 |
| ✓ | | 61.84 | 36.24 | 84.83 | 60.97 |
| | ✓ | 60.88 | 36.99 | 83.95 | 60.61 |
| ✓ | ✓ | **63.43** | **37.36** | **87.12** | **62.63** |

*Table 5.* Performance of adaptive rejection with different threshold factors $w$ under real SalSPML supervision.

| Dataset | $w = 0.5$ | $w = 1$ | $w = 2$ | PiSA |
|---|---|---|---|---|
| COCO | 59.21 | 59.45 | 58.97 | **59.58** |
| VOC | 87.32 | 87.09 | 85.53 | **87.83** |

*Figure 5.* Comparison of annotation bias $\Delta_{\text{bias}}$ on COCO and VOC under real and confidence-based SalSPML settings. PiSA consistently yields the smallest bias across all settings.

bias from 6.22 to 4.49 on COCO and from 4.10 to 3.11 on VOC. These results provide direct evidence that PiSA significantly attenuates the annotation bias introduced by salient supervision, aligning with our theoretical bias-reduction guarantee.

**PiSA improves recognition of non-salient objects.** To examine the effect of prototype guidance, we visualize class activation maps (CAMs) for models trained with and without it. As shown in Figure 7, **without** prototype guidance the model frequently fails to localize visually non-salient instances. In these cases, the target object is weakly expressed (e.g., small, partially occluded, or visually less distinctive), and the model's attention can drift to background regions or co-occurring distractors, producing spurious activation patterns and recognition errors.

In contrast, **with** prototype guidance, the model is encouraged to align its class-wise embeddings with stable category prototypes. This prototype-based reference helps filter out incidental cues and redirects activation toward the true object regions, even when the object is not salient in the scene. Overall, these visualizations indicate that prototype guidance improves localization reliability and is particularly beneficial for recognizing non-salient instances under challenging visual conditions.

We provide a qualitative visualization of the embedding

space learned in Figure 6, which shows that prototype guidance leads to compact and structured embeddings.

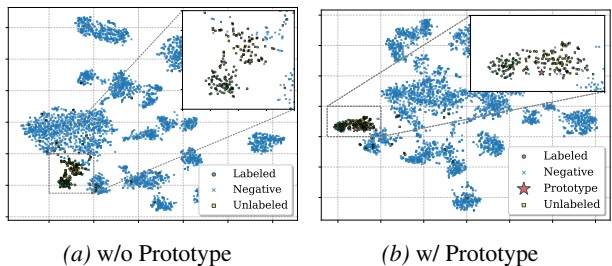

*(a)* w/o Prototype        *(b)* w/ Prototype

*Figure 6.* t-SNE visualization of the embedding space for class 1 (bicycle) on VOC with and without prototype guidance.

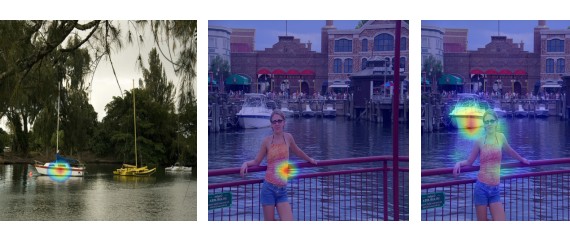

*(a)* salient sample.    *(b)* without PiSA.    *(c)* with PiSA.

*Figure 7.* CAMs are generated from class-wise embeddings for the class boat. Without prototype guidance, the model attends to incorrect regions on non-salient samples. With prototype guidance, PiSA redirects attention to the true object region and recovers correct recognition on non-salient samples.

**PiSA is insensitive to hyperparameters.** In all experiments above, we fix both hyperparameters to $\alpha = \gamma = 1$ throughout training. To further verify the stability of PiSA, we conduct a sensitivity analysis on $\alpha$ and $\gamma$ by varying one parameter while keeping the other fixed.

The parameter $\alpha$ controls the contribution of prototype-guided learning, while $\gamma$ regulates the separation between embeddings of saliently labeled samples and prototypes of other classes. As shown in Figure 8, PiSA achieves its best performance at $\alpha = 1$ on both datasets. Moreover, PiSA maintains consistently strong performance across a wide range of $\gamma$ values in $\{0.1, 0.5, 1, 2\}$, indicating robustness to the choice of this parameter. Results show that PiSA is insensitive to these hyperparameters and does not require extensive tuning in practice.

*Table 6.* Memory and time consumption comparison on VOC with batch size 16.

| Method | Memory (MB) | Time (s) |
|--------|-------------|----------|
| BCE    | 7102        | 44.44    |
| Ours   | 7796        | 60.82    |

**PiSA Remains Efficient.** We compare the computational cost of PiSA with the standard BCE loss on VOC under

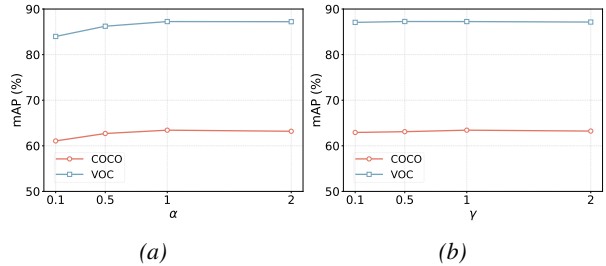

*(a)*                *(b)*

*Figure 8.* Parameter sensitivity of our method PiSA. (a) and (b) represent the mAP of our method on single positive salient label by varying $\alpha$ and $\gamma$ respectively.

the same training setting in Table 6. Due to the class-wise embedding module, PiSA introduces a moderate increase in memory usage and training time per iteration, while remaining practically efficient given the performance gains. Regarding scalability, the prototype update is lightweight because each sample updates at most one class prototype under the SalSPML setting. The main additional cost comes from the class-wise embedding–prototype similarity computation, with complexity $O(BMD)$, where $B$, $M$, and $D$ denote the batch size, number of classes, and embedding dimension, respectively. Thus, PiSA scales similarly to standard mini-batch training with respect to the number of samples, and its extra overhead is primarily class-dependent rather than dataset-size-dependent.

## 5. Conclusion

This paper revisits single positive multi-label learning (SPML) from a more realistic perspective, where annotations are biased toward salient categories. We formalize this setting as SalSPML and construct two realistic datasets, and further evaluate on synthetically constructed SalSPML benchmarks for comparisons. Our theoretical analysis shows that salient supervision induces a persistent annotation bias, leading to a non-vanishing excess risk term. To address SalSPML, we propose PiSA, which leverages the representativeness of salient labels to learn class prototypes that guide class-aware embedding learning and enable prototype-guided loss rejection. Extensive experiments on both real and synthetic benchmarks, together with our theoretical analysis, demonstrate the effectiveness and robustness of PiSA under non-random missing labels across diverse datasets.

**Data and Code Availability** The source code of PiSA and the newly constructed real-world SalSPML datasets are publicly available at https://github.com/RainyWangxy/PiSA.git.

## Acknowledgements

This work was supported by the National Natural Science Foundation of China under Grants U24A20322, 62576094 and 62422118. This work is also supported by Hong Kong UGC under grants UGC/FDS11/E03/24, UGC/FDS11/E03/25, and Hong Kong Research Grants Council under Grant 11219324. This research work is also supported by the Big Data Computing Center of Southeast University.

## Impact Statement

This paper introduces Salient Single-Positive Multi-Label Learning (SalSPML), a more realistic single-positive supervision setting that accounts for salient annotation bias, and proposes an effective strategy that constructs class-wise prototypes from salient-labeled samples to guide recognition of non-salient labels. This paper presents work whose goal is to advance the field of machine learning. There are many potential societal consequences of our work, none of which we feel must be specifically highlighted here.

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

# Table of Contents for the Appendix

## A. Excess Risk Decomposition under Partial Annotation

We consider a hypothesis class $\mathcal{F}$ of predictors $f : \mathcal{X} \to \mathbb{R}^C$. Assume the loss is bounded, i.e., $L(\cdot, \cdot) \in [0, M]$. For a multi-label instance $(x, Y) \sim \mathcal{D}$ with label set $Y \subseteq \{1, \ldots, C\}$, define the population risk

$$R(f) := \mathbb{E}_{(x,Y)\sim\mathcal{D}}\big[L(f(x), Y)\big], \tag{17}$$

and let $f^\star \in \arg\min_{f\in\mathcal{F}} R(f)$ denote a population risk minimizer.

Under partial annotation, the training procedure observes a supervision signal $S$ generated from $Y$ by an annotation process (e.g., Random SPML or SalSPML). Let $\hat{R}_D(f)$ be the empirical risk computed from the observed supervision in the dataset $D = \{(x_i, S_i)\}_{i=1}^n$, i.e.,

$$\hat{R}_D(f) := \frac{1}{n} \sum_{i=1}^n L_{\mathrm{obs}}(f(x_i), S_i), \tag{18}$$

where $L_{\mathrm{obs}}$ is the training loss induced by the observed supervision.

Let $\hat{f} \in \arg\min_{f\in\mathcal{F}} \hat{R}_D(f)$ be the empirical risk minimizer (ERM). The excess risk admits the following decomposition:

$$R(\hat{f}) - R(f^\star) = \underbrace{R(\hat{f}) - \hat{R}_D(\hat{f})}_{\textbf{Generalization Gap}} + \underbrace{\hat{R}_D(\hat{f}) - \hat{R}_D(f^\star)}_{\textbf{Optimization Error}}$$

$$+ \underbrace{\hat{R}_D(f^\star) - R(f^\star)}_{\textbf{Annotative Bias}}. \tag{19}$$

The first term is controlled by standard uniform convergence arguments (e.g., via Rademacher complexity), and vanishes as $n$ grows under mild assumptions. The second term is non-positive for exact ERM; we keep it to allow approximate optimization. The third term, *annotative bias*, captures the systematic discrepancy between the empirical risk under the observed supervision and the true population risk.

In our setting, the key distinction between Random SPML and SalSPML lies in this annotative bias term, as SalSPML induces a non-random, salient selection mechanism that can lead to a persistent bias.

## B. Generalization Bounds (Supplement to Sec. 3.2)

### B.1. Notation

Let $\mathcal{F}$ be the hypothesis class and $L$ a bounded loss: $0 \le L \le M$. For a dataset $D = \{(x_i, s_i)\}_{i=1}^n$, define

$$\hat{R}_D(f) = \frac{1}{n} \sum_{i=1}^n L(f(x_i), s_i), \qquad R(f) = \mathbb{E}_{(x,Y)\sim D}[L_{\mathrm{multi}}(f(x), Y)].$$

We analyze the uniform deviation

$$\Phi(D) = \sup_{f\in\mathcal{F}} \big(R(f) - \hat{R}_D(f)\big).$$

### B.2. McDiarmid Lemma for Uniform Deviation

Changing one sample $(x_i, s_i)$ affects $\Phi(D)$ by at most $M/n$. By McDiarmid inequality, for any $\delta \in (0, 1)$,

$$\Phi(D) \le \mathbb{E}[\Phi(D)] + M\sqrt{\frac{\log(1/\delta)}{2n}} \quad \text{w.p. } \ge 1 - \delta. \tag{20}$$

Thus it remains to bound $\mathbb{E}[\Phi(D)]$.

## B.3. Rademacher Complexity Upper Bound

Introduce i.i.d. Rademacher variables $\epsilon_{ij} \in \{-1, +1\}$. By standard symmetrization,

$$
\begin{aligned}
\mathbb{E}[\Phi(D)] &= \mathbb{E}\left[ \sup_{f \in \mathcal{F}} \left( R(f) - \hat{R}_D(f) \right) \right] \\
&\leq 2\,\mathbb{E}\left[ \sup_{f \in \mathcal{F}} \frac{1}{n} \sum_{i=1}^{n} \sum_{j=1}^{C} \epsilon_{ij} L(f(x_i), j) \right] \\
&= 2\,\mathbb{E}\left[ \hat{\mathfrak{R}}_n(L \circ \mathcal{F}) \right].
\end{aligned}
$$

Combining with Eq. (20), we obtain

$$
\sup_{f \in \mathcal{F}} |R(f) - \hat{R}_D(f)| \leq 2\mathbb{E}\left[ \hat{\mathfrak{R}}_n(L \circ \mathcal{F}) \right] + M\sqrt{\frac{\log(1/\delta)}{2n}}. \tag{21}
$$

This confirms that the first two terms in Eq. (19) vanish as $n \to \infty$ at a rate of $\mathcal{O}(1/\sqrt{n})$, leaving $\Delta_{\text{bias}}$ as the dominant term in SalSPML.

## C. Proof of Theorem 3.2 (Hardness of SalSPML)

We analyze the bias term $\Delta_{\text{bias}} = \hat{R}_D(f^\star) - R(f^\star)$ under Random and Salient selection settings.

**Random Selection (Unbiased).** Let $Y$ be the set of ground-truth positive labels for input $x$. In Random SPML, the observed label $s$ is drawn uniformly from $Y$, i.e., $P(s = j|x, Y) = 1/|Y|$ for all $j \in Y$. The expected loss for a sample is:

$$
\mathbb{E}_s[L(f(x), s)] = \sum_{j \in Y} \frac{1}{|Y|} L(f(x), j) = \frac{1}{|Y|} L(f(x), Y). \tag{22}
$$

Since the expected empirical risk is proportional to the true risk (up to the scaling factor $1/|Y|$), the optimization is statistically consistent. Thus, $\Delta_{\text{bias}} = 0$.

**Salient Selection (Biased).** In SalSPML, the selection is not uniform. Let $s^{\text{sal}}$ be the salient label and $Y^{\text{hard}} = Y \setminus \{s^{\text{sal}}\}$ be the set of unobserved hard positives. The true population risk over all positives is:

$$
R(f) = \mathbb{E}\left[ L(f(x), s^{\text{sal}}) + \sum_{j \in Y^{\text{hard}}} L(f(x), j) \right]. \tag{23}
$$

However, the empirical risk only observes $s^{\text{sal}}$:

$$
\mathbb{E}[\hat{R}_{\text{sal}}(f)] = \mathbb{E}[L(f(x), s^{\text{sal}})]. \tag{24}
$$

Subtracting the two yields the irreducible bias:

$$
\Delta_{\text{bias}} = R(f) - \mathbb{E}[\hat{R}_{\text{sal}}(f)] = \mathbb{E}_{(x,Y)}\left[ \sum_{j \in Y^{\text{hard}}} L(f(x), j) \right]. \tag{25}
$$

Since the loss is non-negative and $Y^{\text{hard}}$ contains valid positives that the model fails to predict if trained only on $s^{\text{sal}}$, $\Delta_{\text{bias}} > 0$. As $n \to \infty$, the ERM converges to a solution that minimizes loss on salient labels but ignores hard labels, proving the theorem.

## D. Proof of Theorem 3.3 (Bias Reduction)

Let $z = f_{\text{enc}}(x) \in \mathbb{R}^D$ be the encoded feature and $\mu_j = \mathbb{E}[z \mid j \in Y]$ the true class mean. Assume the learned prototype $p_j$ satisfies $\|p_j - \mu_j\| \leq \epsilon$. Define the similarity score $s_j(z) = \langle z, p_j \rangle$ and the threshold $\tau_j = \|\mu_j\|^2 - \gamma$ with $\gamma > 0$.

**Recovery probability.** For any true positive label $j$, write $z = \mu_j + \xi$ where $\xi$ is zero-mean sub-Gaussian noise (with parameter $\sigma$). Then $s_j(z) = \langle \mu_j, p_j \rangle + \langle \xi, p_j \rangle$. Therefore, the failure event can be rewritten as

$$\Pr\big(s_j(z) < \tau_j\big) = \Pr\Big(\langle \xi, p_j \rangle < \tau_j - \langle \mu_j, p_j \rangle\Big) = \Pr\Big(\langle \xi, p_j \rangle < \langle \mu_j, \mu_j - p_j \rangle - \gamma\Big). \tag{26}$$

Next, by Cauchy–Schwarz and $\|p_j - \mu_j\| \le \epsilon$, we have $\langle \mu_j, \mu_j - p_j \rangle \le \|\mu_j\| \|\mu_j - p_j\| \le \epsilon \|\mu_j\|$. Plugging this into Eq. (26) gives $\Pr(s_j(z) < \tau_j) \le \Pr(\langle \xi, p_j \rangle < -(\gamma - \epsilon \|\mu_j\|))$. Since $\langle \xi, p_j \rangle$ is sub-Gaussian with parameter $\sigma \|p_j\|$, its tail satisfies

$$\Pr\big(\langle \xi, p_j \rangle \le -t\big) \le \exp\Big(-\frac{t^2}{2\sigma^2 \|p_j\|^2}\Big), \quad t > 0. \tag{27}$$

Taking $t = \gamma - \epsilon \|\mu_j\|$ (thus requiring $\gamma > \epsilon \|\mu_j\|$), we obtain

$$\eta = 1 - \Pr\big(s_j(z) < \tau_j\big) \ge 1 - \exp\Big(-\frac{(\gamma - \epsilon \|\mu_j\|)^2}{2\sigma^2 \|p_j\|^2}\Big) = 1 - \exp\big(-\Omega((\gamma - \epsilon)^2)\big), \tag{28}$$

where the last equality follows under standard normalization (e.g., bounded $\|\mu_j\|$ and $\|p_j\|$), which is absorbed into constants.

**Impact on the bias term.** Let $Y^{\text{hard}}$ be the missed positives under salient supervision. If each hard positive is recovered independently with probability at least $\eta$, then an expected fraction $(1 - \eta)$ remains unrecovered, and the bias term reduces to

$$\Delta_{\text{bias}}^{\text{new}} = (1 - \eta)\Delta_{\text{bias}}. \tag{29}$$

Substituting this into Eq. (21) completes the proof of Theorem 3.3.

# E. Performance under CLIP-based Datasets

We leverage a pretrained vision–language model (CLIP) to simulate human saliency judgments. For each image, the label with the highest CLIP prediction confidence is retained as the single positive label, while all other labels are treated as unknown. Notably, our method yields consistent gains on VOC (89.47) and COCO (70.94). On NUS-WIDE, our method is slightly below the best result (CS: 47.69), which may be attributed to the larger label space and higher semantic correlation among labels in NUS, where CLIP-based selection can introduce more ambiguity/noise (e.g., class bias). Overall, our method achieves the best average performance.

*Table 7.* Experimental results of our method on three benchmark datasets under the CLIP-based SalSPML setting. The metric reported is mAP. Bold font indicates the best performance.

| Method | VOC | COCO | NUS | Avg. | Sig. |
|---|---|---|---|---|---|
| BCE | 86.04±0.22 | 62.64±0.31 | 43.64±0.47 | 64.11 | 3 / 0 / 0 |
| ASL | 87.13±0.26 | 64.06±0.17 | 44.53±0.68 | 65.24 | 2 / 1 / 0 |
| ROLE | 87.37±0.15 | 61.29±0.17 | 39.73±0.25 | 62.80 | 3 / 0 / 0 |
| LL-Ct | 86.18±0.08 | 65.20±0.06 | 46.77±0.37 | 66.05 | 2 / 1 / 0 |
| LL-R | 87.19±0.26 | 65.50±0.03 | 46.59±0.22 | 66.43 | 2 / 1 / 0 |
| EM | 87.82±0.16 | 64.66±0.12 | 42.85±0.16 | 65.11 | 3 / 0 / 0 |
| ML-Decoder | 87.67±0.45 | 69.53±0.48 | 45.11±0.32 | 67.44 | 2 / 1 / 0 |
| BoostLU | 87.37±0.21 | 66.35±0.17 | 46.82±0.29 | 66.85 | 2 / 1 / 0 |
| GRLoss | 87.75±0.15 | 64.30±0.24 | 44.22±0.10 | 65.42 | 3 / 0 / 0 |
| CS | 87.21±0.19 | 66.14±0.20 | **47.69±0.09** | 67.01 | 2 / 0 / 1 |
| Ours | **89.47±0.19** | **70.94±0.05** | 46.33±0.36 | **68.91** | 24 / 5 / 1 |

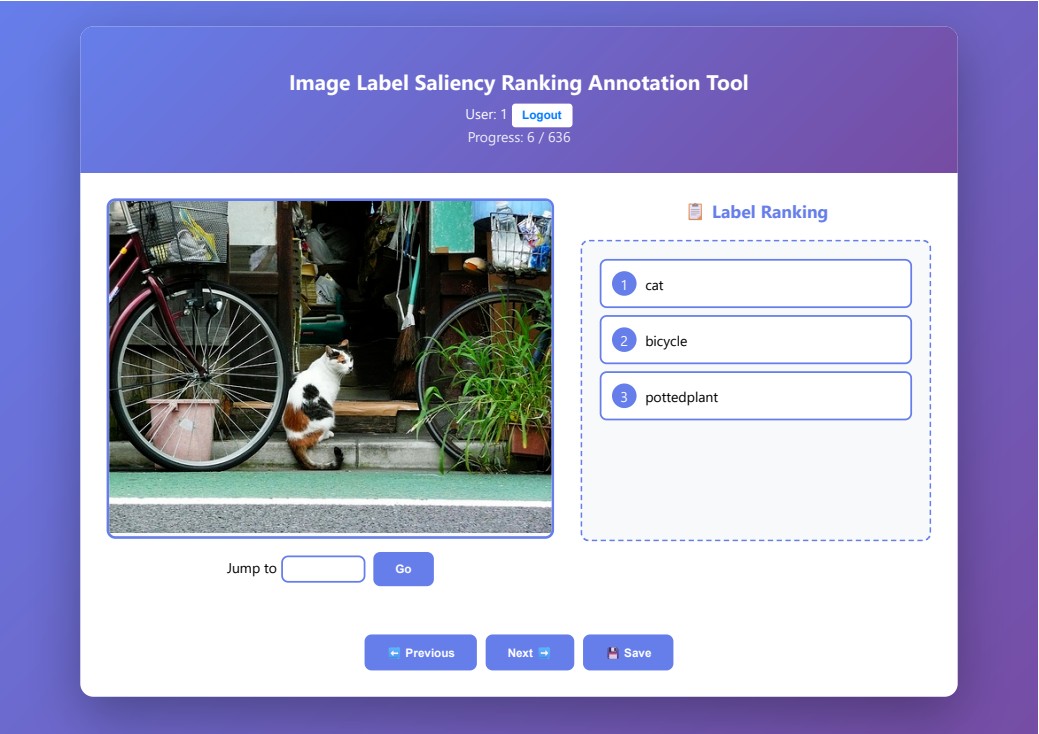

*Figure 9.* The annotation interface used for salient label ranking. Annotators are asked to rank all positive labels of an image according to visual saliency and retain only the most salient label as supervision.

## F. Annotation Interface

Figure 9 illustrates the annotation interface used to construct the real-world SalSPML datasets. For each image, annotators are presented with the image on the left and a list of its ground-truth positive labels on the right. They are instructed to rank these labels based on visual saliency, i.e., how noticeable or prominent each category appears in the image. Only the top-ranked label is retained as the salient single-positive annotation, while the remaining labels are treated as unobserved. The interface supports sequential navigation and progress tracking to ensure annotation consistency and efficiency. Figure 10 further illustrates the difference between SalSPML and conventional SPML through representative examples.

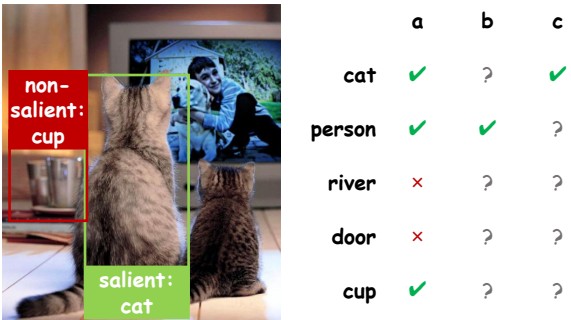

*Figure 10.* Example of instance with ground-truth labels (a), single positive label (b), and salient single positive label (c). Instances in (c) are annotated with a subset of positive labels that are easier to attend to than those in (b).

