# OpenReview forum: "One Coin Has Two Sides: Single Poistive Multi Label Learning from Salient Annotations"
_ICML.cc/2026/Conference — ICML 2026 regular_

### Official Review · Reviewer_9JVr · 2026-02-24

**Soundness:** 3
**Presentation:** 3
**Significance:** 2
**Originality:** 3
**Overall Recommendation:** 5
**Confidence:** 4

**Summary:**

This paper introduces a new multi-label learning setting termed SalSPML, which aims to better reflect real-world annotation behaviors where annotators tend to label only the most visually salient object in an image. To address the inherent annotation bias in this setting, the authors propose a prototype-guided method called PiSA. This approach constructs class prototypes and aligns sample embeddings with them, while also incorporating a prototype-guided loss rejection mechanism to mitigate the impact of false negative labels. The method is validated on both human-annotated and synthetic datasets, demonstrating its superiority over existing baselines. Theoretical analysis is also provided to characterize the bias and its reduction.

**Compliance With Llm Reviewing Policy:**

Affirmed.

**Final Justification:**

All my concerns have been addressed.

**Key Questions For Authors:**

See the above weaknesses.

**Limitations:**

No. The authors have not introduced any limitations of this paper.

**Strengths And Weaknesses:**

***Strengths:***

- The proposed SalSPML setting is more realistic than the conventional random single-positive assumption.

- The paper includes extensive experiments, comprehensive ablation studies, sensitivity analyses, which jointly support the effectiveness of the proposed method.

- The authors provide a theoretical decomposition of the annotation bias and a proof of its reducibility.

***Weaknesses:***

- Since prototypes are updated from biased salient samples, how sensitive is the method to the initial warm-up phase? Is a single epoch sufficient to obtain reliable prototypes, and could a more robust initialization strategy (e.g., using a pretrained model) further improve performance?

- The rejection criterion in Eq. (13) uses a fixed top-k threshold. Is this choice (k=3) dataset-dependent? Is there a principled way to determine or adapt this parameter dynamically during training?

- Theorem 3.3 assumes a margin condition that true positives are more similar to their matched prototypes than to mismatched ones. Is this assumption valid for non-salient hard positives under significant annotation bias? If the condition is not met, does the recovery probability guarantee still hold?

- Several compared methods (e.g., CS, GRLoss) were not originally designed for the SalSPML setting. Were any modifications made to adapt these baselines to this new task? If not, this may affect the fairness of the comparison.

- While memory and time comparisons are provided in the appendix, the paper does not discuss the scalability of the prototype updates and similarity computations on larger datasets.

---

> ### Author Rebuttal · Authors · 2026-03-31
>
> We sincerely thank the Reviewer for the rigorous evaluation and insightful comments, which help improve the clarity and depth of our manuscript.
> ## W1. PiSA is not Sensitive to Warm-up, and 1 Epoch is Stable
> A short warm-up is common in related SPML methods **[1,2]**.
> Moreover, **PiSA does not keep warm-up prototypes fixed**: they only provide an initial anchor and are updated online.
>
> To examine warm-up sensitivity, we vary the number of warm-up epochs in {0,1,3,5} and report the results in **Table R1**.
> Since all experiments use a pretrained ResNet-50, **warm-up=0 means using pretrained features to construct prototypes.**
>
> **Table R1.** Effect of Warm-up Epochs on PiSA Performance (mAP).
> |Dataset/warm-up epochs|0|1|3|5|
> |-|-|-|-|-|
> |COCO|58.76|**59.58**|59.34|58.62|
> |Pascal|87.49|**87.83**|86.12|85.57|
>
> As shown in **Table R1**, PiSA is robust to warm-up length. In particular, **1 epoch gives strong and stable performance, while longer warm-up does not consistently and can even hurt**, especially on Pascal.
> To avoid overfitting caused by excessive warm-up, we use 1 epoch as a trade-off.
> ## W2. Fixed Threshold $k$ is Robust, and Adaptive Rejection Behaves Similarly
> Combined with **Fig. 4**, our results show that fixed $k=3$ is stable. This choice is motivated by the strong positive-negative imbalance in multi-label learning: **rejecting a few extra negatives usually has limited effect, while keeping false negatives in supervision is much more harmful.**
>
> According to your comment, we test an adaptive rule: for each image, we compute the mean $\mu$ and standard deviation $\sigma$ of similarities to all class prototypes, and mask class $c$ if its similarity exceeds $\mu+w\sigma$. If more than 3 classes satisfy the rule, we keep only the top-3.Results are shown in **Table R2** and **Table R3.**.
>
> **Table R2.** Performance of adaptive rejection with different threshold factors $w$  and PiSA under REAL supervision (mAP).
> |Dataset/Method|$w=0.5$|$w=1$|$w=2$|PiSA|
> |-|-|-|-|-|
> |COCO|59.21|59.45|58.97|**59.58**|
> |Pascal|87.32|87.09|85.53|**87.83**|
>
> **Table R3.** Distribution of rejected classes per image on COCO under adaptive rejection.
> |Rejected classes/w|0.5|1|2|
> |-|-| -|-|
> |0|0.0%|0.0%|51.9%|
> |1|0.0%|0.9%|36.8%|
> |2|86.9%|80.3%|10.4%|
> |3|13.1%|18.8%|0.9%|
>
> In **Table R2**, adaptive rejection's performance is close to PiSA.
> In **Table R3**, when $w=0.5$ or $1$, the adaptive rule rejects 2 Classes for most images.
> So we choose fixed threshold (k=3>2) as a robust strategy.
>
> ## W3. **Theorem 3.3** is Conditional, but its Key Assumption is still Supported in Practice
> We agree that **Theorem 3.3** is a conditional guarantee. Its recovery bound requires both accurate prototypes ($|p_j-\mu_j|\le\epsilon$) and a sufficiently positive matched-vs.-mismatched margin; Appendix D further requires $\gamma > \epsilon|\mu_j|$. If this is severely violated for extreme non-salient hard positives, the strict guarantee no longer directly applies.
>
> However **theoretically**, **Theorem 3.3 does not require hard positives to share the same appearance distribution as salient ones.**
> It only requires enough class-level semantic consistency for true positives to remain relatively closer to their matched prototype.
> **Empirically**, the t-SNE visualization in **Appendix E.3** suggest that this margin remains acceptable.
> Consistently, as discussed in our response to **Reviewer nE8V-q's I.2 & II.1**, **PiSA preserves clear recognition ability on hard samples**.
> So PiSA uses class-wise embeddings rather than a single global image feature.
>
> ## W4. Baselines were Compared Fairly under the Same SalSPML Setup
> **Our goal is to evaluate how existing SPML methods perform under SalSPML, rather than redesign each baseline for this new problem.**
> We did not modify the core algorithms of compared methods such as CS and GRLoss. All baselines were retrained under the same SalSPML supervision, with the same train/validation split, ImageNet-pretrained ResNet-50 backbone. So, **the comparison is fair**.
> Details are in **Section 4.2**.
>
> ## W5. PiSA Adds Modest Class-Dependent Overhead beyond Standard Training
> In practice, the dominant cost still comes from the backbone forward/backward passes, as in standard multi-label training. The prototype update is lightweight, since under SalSPML each sample updates at most one class prototype. The main extra cost comes from the class-wise embedding–prototype similarity computation in **Eqs. (12)–(14)**, whose complexity is $O(BCD)$, where $B$ is the batch size, $C$ the number of classes, and $D$ the embedding dimension. Therefore, PiSA scales similarly to standard mini-batch training with respect to the number of samples, while its additional overhead is mainly **class-dependent** rather than dataset size.
>
> **[1]** Large loss matters in weakly supervised multi-label classification, CVPR 2022.
>
> **[2]** Bridging the gap between model explanations in partially annotated multi-label classification, CVPR 2023.

---

> > ### Author Rebuttal · Reviewer_9JVr · 2026-04-01
> >
> > All my concerns have been addressed and I support to accept this paper.

---

> > > ### Author Response · Authors · 2026-04-01
> > >
> > > Dear Reviwer **9JVr**,
> > >
> > > We are happy that your concerns have been adequately addressed. Thanks for raising the score. Thanks again for your time and effort in reviewing this paper.
> > >
> > > Regards from the authors.

---

### Official Review · Reviewer_bpgB · 2026-03-09

**Soundness:** 3
**Presentation:** 3
**Significance:** 3
**Originality:** 3
**Overall Recommendation:** 5
**Confidence:** 5

**Summary:**

This paper studies the problem of single-positive multi-label learning under the salient annotation setting. In this setting, annotators tend to label only the most salient label rather than randomly selecting one from all true labels. To address this issue, the authors propose a prototype-guided learning framework that uses class prototypes to guide model training and to select reliable labels for optimization. The paper theoretically shows that the salient annotation setting introduces annotative bias in the learning process. Furthermore, the authors demonstrate that prototype-based learning can help identify potential missing positive labels and thus reduce the annotation bias.

**Compliance With Llm Reviewing Policy:**

Affirmed.

**Final Justification:**

The responses have addressed my concerns. I keep the score unchanged.

**Key Questions For Authors:**

1.	During the prototype-guided label rejection stage, why are the labels that are likely to be potential positive labels not directly assigned as positive labels and used for training?

2.	The appendix mentions that a dataset was obtained using an annotation tool. Was this dataset used in the experiments reported in the paper? It seems that the Salient Annotations dataset used in the main text was not manually annotated.

**Limitations:**

The authors could discuss the applicability of the proposed method, for example, explaining why commonly used multi-label learning datasets such as CUB are not included in the experiments.

**Strengths And Weaknesses:**

Strengths：

1.	The research problem addressed in this paper is well motivated and closely aligned with real-world scenarios, making the study practically meaningful.

2.	The paper is clearly organized, and the proposed method is well aligned with the motivation of the work.

3.	The experimental evaluation is comprehensive. The authors conduct extensive experiments to validate the proposed approach, and the method achieves strong empirical performance.

Weaknesses

1.	The results shown in Figure 1 are somewhat unclear. For the data under the rand setting, since the true number of positive labels is unknown during training, probability correction cannot be applied, which also leads to a biased setting. Therefore, both settings are biased. However, the large performance gap observed between them is not well explained. As a result, it is difficult to clearly attribute the improvement to the proposed prototype-guided bias reduction.

2.	Compared with the baseline methods, the proposed framework introduces an additional class-wise encoder. The impact of this additional encoder on the model performance should be further investigated and discussed.

---

> ### Author Rebuttal · Authors · 2026-03-31
>
> We sincerely thank the Reviewer-bpgB for the supportive and constructive comments. Below we address each point in detail, and we will incorporate these clarifications into the revised manuscript.
>
> ## W1. On the Gap Between Random and Salient Settings
> Our main point is not that Random SPML is completely unbiased, but rather that previous SPML methods are generally designed under the assumption that **the observed single positive label is selected randomly from the set of true positives.**
> However, in reality, the retained label tends to be the most **visually salient** and easily recognizable positive, while unsalient positives are consistently suppressed.
> **This discrepancy is precisely the annotation bias formalized in Section 3.2.**
> **Fig. 1** is intended to illustrate that when salient annotations violate the random-SPML assumption, methods built upon that assumption can suffer substantial performance degradation.
>
> ## W2. Class-wise-embedding's Discussion
> The Class-wise-embedding encoder is part of the underlying architecture of our method. Under the SalSPML setting, a single global embedding can be overly biased toward salient objects, thus weakening the representation of other potential positives.
> By introducing the class-wise embedding encoder, the model generates a dedicated embedding for each class, thereby reducing the dominance of salient objects and improving overall performance.
>
> **Table R1.** Effect of the class-wise embedding encoder and the full PiSA model (mAP, mean ± std). Here, PiSA-E denotes the model with the class-wise embedding encoder trained only with BCE, without PL or PGLR.
> | Method | COCO | VOC | Average |
> |:-------|:----:|:---:|:---:|
> | PiSA-E | 56.11 $\pm$ 0.29 | 83.77 $\pm$ 0.75 | 69.94 |
> | PiSA   | 59.58 $\pm$ 0.29 | 87.83 $\pm$ 0.45 | 73.71 |
>
> As shown in **Table R1**, adding the class-wise embedding encoder brings improvement,confirming the effectiveness of the encoder itself.
> However, the full PiSA model performs substantially better, indicating that the gains also come from PL and PGLR.
>
> We will clarify this architectural contribution and its trade-off more explicitly in the revised manuscript.
>
> ## Q1. Why We Do Not Directly Assign Missing Labels as Positives
> Our proposed model intentionally adopts a more conservative strategy.
> Under the salient setting, unsalient positives are inherently harder to identify accurately and error correction is more prone to mistakes.
> Instead of directly assigning potentially missing labels as positives, **we use a prototype-guided rejection mechanism to filter out loss terms that are likely to correspond to false negatives, thereby avoiding incorrect negative supervision.**
> In this case, we use a relatively loose threshold for loss rejection rather than aggressively introducing additional positive supervision, **because incorrectly introducing a false-positive supervision signal is often more harmful than conservatively rejecting a portion of unreliable losses**.
>
> ## Q2. On the Use of Real Salient Datasets
> Yes.
> The results reported in **Table 1** in our paper are based on our manually annotated real salient-label datasets. As described in **Section 4.1**, we recruited 10 annotators, each of whom selected the single  most visually salient object in an image as the supervision label.
> **Appendix G** further presents the annotation interface used in this process.
> However, we acknowledge that the current presentation may cause confusion between experiments with real salient labels and those with synthetic salient labels, and we will make this distinction clearer in the revised version.
>
> ## L1. On Limitations and Applicability
> Our current experiments mainly focus on widely used multi-label benchmark datasets (including COCO, VOC, NUS-WIDE, and VG), because they naturally involve multiple objects, incomplete labels, and salient/unsalient scenarios. In contrast, CUB is a fine-grained dataset whose labels are primarily associated with attributes or species categories, making its annotation characteristics less aligned with the salient single-positive setting studied in this work. In particular, **under fine-grained recognition scenarios, the notion of the most salient positive label is often less natural.**
> We will add a more explicit discussion of this issue in the revision to better clarify the intended scope and limitations of SalSPML.

---

> > ### Author Rebuttal · Reviewer_bpgB · 2026-04-02
> >
> > I keep my score unchanged.

---

> > > ### Author Response · Authors · 2026-04-04
> > >
> > > Dear Reviewer **bpgB**,
> > >
> > > Thanks for your reply. **We are happy that your concerns have been fully resolved**. Thanks again for your time and effort in handling this paper.
> > >
> > > Regards from the authors.

---

### Official Review · Reviewer_nE8V · 2026-03-11

**Soundness:** 2
**Presentation:** 3
**Significance:** 3
**Originality:** 2
**Overall Recommendation:** 4
**Confidence:** 3

**Summary:**

This paper addresses Salient Single-Positive Multi-Label Learning (SalSPML), where annotating only the most visually prominent label introduces a systematic bias. To tackle this, the paper proposes PiSA, a framework that leverages highly representative salient annotations to build reliable class-wise prototypes for supervising embedding learning. Furthermore, the paper introduces a Prototype-Guided Loss Rejection (PGLR) mechanism to adaptively suppress false-negative signals based on feature-prototype similarities. Extensive experiments on synthetic benchmarks and two newly constructed real-world datasets demonstrate that PiSA effectively mitigates annotation bias and achieves competitive performance.

**Compliance With Llm Reviewing Policy:**

Affirmed.

**Final Justification:**

The rebuttal and subsequent discussion have addressed my main concerns: distribution shift and prototype validity are supported by hard-sample experiments and prototype-to-sample distance analysis; Theorem 3.3's practical applicability is clarified; inter-annotator agreement confirms dataset reliability; the stochastic generalization of Theorem 3.2 shows the bias bound holds under probabilistic annotation; the MNAR/PU distinction is better argued both theoretically and empirically; and failure case analysis is provided. Overall satisfactory. I raise my score and encourage the authors to incorporate all the above clarifications and additional results into the revised manuscript.

**Key Questions For Authors:**

See weaknesses.

**Limitations:**

Not explicitly discussed. Would be good to include a clear discussion on this.

**Strengths And Weaknesses:**

Strengths:
- The paper investigates Single Positive Multi-label Learning (SPML), which is a highly relevant and practical problem in machine learning given the prohibitively high cost of exhaustive annotations in real-world scenarios.
- Empirical results confirm the efficacy of the proposed PiSA framework. The ablation studies validate the utility of the proposed modules, and the overall method successfully outperforms previous SPML baselines on the newly constructed datasets.
- The paper is well-written and easy to follow.

Weaknesses:

I. Motivation
1. The setting is overly absolute and deviates from real-world complexity.
The paper formalizes salient annotation in Equation (3) as a deterministic function $y^{sal}=\phi(x,Y)$. However, real-world human annotation behavior is highly stochastic and noisy. Annotators are influenced not only by visual saliency but also by "Center Bias" (tending to annotate objects in the center of the image), "Familiarity Bias" (tending to annotate categories they are most familiar with), and even annotation fatigue. Forcing a complex human behavior, which is essentially a stochastic process, into a deterministic function that always returns the "most salient label" not only oversimplifies the problem but also weakens the generalization ability of its theoretical derivations (Theorem 3.2) in real-world scenarios. I am interested in seeing a discussion on how the theoretical bounds of $\Delta_{bias}$ would change if the annotation process were not purely deterministic (e.g., saliency sampling with a certain probability distribution).

2. The "Two Sides of One Coin" hypothesis suffers from intrinsic Distribution Shift.
The paper hypothesizes that the features of salient labels can serve as high-quality semantic anchors. However, in the visual representation space, a "salient cat" occupying 80% of the image with perfect lighting and a "non-salient cat" that is 50% occluded in the dark often exhibit a massive Distribution Shift. The proposed method forcibly pulls the features of these non-salient samples toward the feature prototypes dominated by salient samples via $\mathcal{L}_{PL}$. This approach risks obliterating the unique, long-tail features of non-salient samples caused by occlusion, small object size, etc., leading to a catastrophic forgetting of the model's ability to recognize hard examples. It would be highly beneficial if the paper could provide visualization or quantitative analysis of the prototype space to demonstrate that forced alignment does not destroy the manifold structure of non-salient features. Furthermore, I am very interested in whether the paper could conduct quantitative experiments on benchmarks specifically designed to highlight hard examples (e.g., datasets focusing on heavy occlusion, small objects, or long-tail distributions). Evaluating the method strictly on these challenging subsets would provide much stronger empirical evidence that the prototype alignment does not compromise the recognition of non-salient, difficult instances.
3. SalSPML is essentially an instance of MNAR; the conceptual novelty is over-packaged.
The paper positions "Salient Single-Positive Multi-Label Learning (SalSPML)" as a novel and more realistic problem setting. However, stripping away the application-specific terminology of "saliency", this formulation is essentially a specific instance of Positive-Unlabeled (PU) learning with Selection Bias, or learning under the Selected Not At Random (SNAR / MNAR) assumption, a topic that has been extensively formalized and studied in the core machine learning community. In conventional PU and Single-Positive Multi-Label (SPML) learning, methods often rely on the Selected Completely At Random (SCAR) assumption. The paper argues that in reality, annotators favor "visually prominent" objects, meaning the probability of observing a positive label is highly dependent on the features of the instance itself. This exact phenomenon has already been well-documented in top-tier ML venues:
Kato et al. [1] explicitly criticized the SCAR assumption, formalizing the setting where positive labels are subject to severe selection bias (e.g., clear/prominent instances are labeled, while ambiguous ones are left unlabeled);
Hammoudeh et al. [2] introduced the bPU (biased-PU) framework to handle scenarios where the labeled positive set is merely a biased subset of the target distribution;
Furthermore, Bekker and Davis [3] established the theoretical taxonomy for this, defining it strictly as the SNAR (Selected Not At Random) mechanism, which is mathematically equivalent to Missing Not At Random (MNAR) in missing data theory.
Recent advances (e.g., PUe [4], NeurIPS 2023) have even adopted causal inference to correct this exact feature-dependent observation bias.
The current manuscript rebrands this well-established selection bias problem as "SalSPML", yet fails to acknowledge or theoretically compare its approach against the rich lineage of biased-PU (bPU) and SNAR learning literature. I am very interested in seeing the paper properly contextualize this setting within the broader ML literature on selection bias, and critically discuss why the prototype-based alignment is theoretically or empirically superior to existing unbiased risk estimators or propensity-scoring methods designed specifically for non-random label missingness.


II. Methodology
1. A logical gap exists in the core assumption of the theoretical proof (Theorem 3.3).
In the proof of Theorem 3.3, the paper makes a highly critical assumption: the distance between the learned prototype $P_c$ and the true class mean $\mu_c$ is bounded, i.e., $||P_c-\mu_c||\le\epsilon$. This appears to be a logical flaw. $\mu_c$ is the true distribution mean based on all positive samples (both salient and non-salient). However, in the algorithm's execution, the prototypes are generated solely by aggregating the features of salient samples (Eq. 8). Because there is a severe distribution shift between salient and non-salient samples, the $P_c$ generated from locally biased data cannot be guaranteed to unbiasedly approximate the global true mean $\mu_c$. If $\epsilon$ is actually very large, the recovery probability $\eta$ derived in Equation (27) will be extremely low, and the purported "bias mitigation" theoretical guarantee collapses. I would appreciate it if the paper could provide rigorous proofs or empirical evidence explaining why a prototype constructed only from biased salient samples can approximate the true mean of all samples with a small $\epsilon$.

2. The fixed hyperparameter $k$ results in a lack of adaptability in the rejection strategy.
In Prototype-Guided Loss Rejection (PGLR), the method rejects false negatives through hard truncation, discarding the top-$k$ unlabeled categories with the highest similarity (fixed at $k=3$ in experiments). This hard-coding could be somewhat fragile. In the COCO dataset, an image might contain objects from over a dozen different categories; whereas in the VOC dataset, many images contain only a single object. If an image intrinsically contains only 1 object, forcibly setting $k=3$ means the model will mistakenly discard 2 True Negatives as false negatives, needlessly losing valuable negative supervision signals. This approach seems to lack consideration for dynamic thresholds. I am interested in whether the paper could explore rejection strategies based on adaptive thresholds (e.g., based on the variance of the similarity distribution, rather than a fixed Top-k), or conduct fine-grained analysis on images with different object densities.

3. Confirmation Bias introduced by Phase I Warm-up.
The model undergoes a 1-epoch warm-up using the BCE loss in the first phase, during which feature prototypes are updated. Because the supervision signal is entirely dominated by salient labels, the model will inevitably rapidly overfit to salient features during this epoch. Subsequently, the algorithm directly uses these already severely biased prototypes to guide feature alignment and loss rejection in the second phase. This creates a negative feedback loop (Confirmation Bias): using biased features to extract biased prototypes, and then using biased prototypes to filter labels, which might instead lock the model's attention rigidly onto salient objects. I am curious to know how the paper prevents prototype collapse or excessive bias during the early stages, and whether there is experimental evidence showing the drift of prototypes during the first phase.

III. Experimental Results
1. Insufficient Baseline Comparisons.
It would be highly valuable to compare the results with benchmarks from prior machine learning literature related to this topic, particularly methods addressing the Missing Not At Random (MNAR) problem. Theoretically, the multi-label resolution strategy proposed in the paper should also be applicable to binary classification problems under similar selection biases.
2. The construction of the real-world dataset lacks support from authoritative evaluation metrics.
The paper mentions recruiting 10 annotators to construct the real SalSPML datasets. However, much like the dilemma faced when evaluating open-source projects lacking authoritative datasets, "saliency" is a highly subjective concept. The paper merely states "only retain the most salient label," without providing any annotator consistency tests (e.g., Fleiss' Kappa score). If one annotator considers the cat in the foreground to be salient, while another considers a brightly colored bicycle to be salient, this high degree of subjective disagreement will result in a test set full of noise, rendering it challenging to act as an authoritative benchmark for judging model performance. I would love to see the paper provide statistical analysis of the constructed real-world datasets, particularly quantitative metrics of Inter-annotator agreement, to further demonstrate the authority and reliability of the test set.

3. Qualitative Analysis is too thin and lacks Failure Case analysis.
In Figure 6, the paper only presents a successful CAM visualization case for a "boat". For a top-tier conference paper, this might not be sufficiently rigorous. If the model's false-negative rejection mechanism is overly aggressive, could it lead to catastrophic forgetting, mistakenly rejecting originally correct non-salient classes? I am very interested in seeing a failure case analysis added to the paper. For example, how does PiSA's prototype guidance mechanism fail in scenarios where objects are extremely dense, or when a salient prototype (like "person") covers a vastly broad semantic space? Showcasing the model's imperfections will actually enhance the academic rigor of the paper.

[1] Learning from Positive and Unlabeled Data with a Selection Bias. ICLR 2019

[2] Learning from Positive and Unlabeled Data with Arbitrary Positive Shift. NeurIPS 2020

[3] Beyond the Selected Completely At Random Assumption for Learning from Positive and Unlabeled Data. ECML PKDD 2019

[4] PUe: Biased Positive-Unlabeled Learning Enhancement by Causal Inference. NeurIPS 2023

---

> ### Author Rebuttal · Authors · 2026-03-31
>
> We thank the reviewer for their careful reading and constructive feedback.
> ## I.1 Deterministic Saliency Formulation
> We do not claim real human annotation is deterministic. **Eq. (3)** is a simplified abstraction of the key difference from prior SPML: **the observed positive is not randomly annotated from the true label set, but systematically biased toward salient labels.**
> Additionally, our experiments are not limited to a deterministic simulator. We build real-world SalSPML benchmarks with natural human variability, and we test multiple salient-label construction strategies (Confidence-, Size-, and CLIP-based), with PiSA remaining consistently effective.
> ## I.2&II.1 Prototype Alignment and Validity
> PiSA does not force unlabeled embeddings to align with prototypes. $L_{PL}$ is **applied only to salient labeled embeddings** to improve class discrimination, while unlabeled embeddings are used only in prototype-guided rejection to exclude likely false negatives from negative supervision.
>
> Likewise, **Theorem 3.3 does not claim prototypes built from salient samples are unbiased estimates of the full class distribution.**
> Instead, it shows that recovery is still guaranteed as long as the prototype error remains bounded.
>
> Besides the t-SNE feature visualization (see **Fig. 7**), we also quantitatively test whether prototypes learned from salient samples remain informative for unsalient sample.
> We first compute the distance from each class prototype to salient positives, hard positives, and negatives on the validation set.
> The results are:
> $d(P,\mu_{all})=0.009$,
> $d(P,\mu_{sal})=0.007$,
> $d(P,\mu_{hard})=0.016$,
> $d(P,\mu_{neg})=0.033$, where $d(·,·)$ denotes mean cosine distance.
> Although hard positives are farther than salient positives, they are closer than negatives.
>
> We further **define objects with area ratio below 1% as hard samples** and evaluate performance on this subset. As shown in **Table R1**, PiSA consistently outperforms other baselines:
>
> **Table R1.** Performance on hard samples under different SalSPML settings (mAP).
> ||REAL|SIZE|CONF|
> |-|-|-|-|
> |BCE|44.49|41.77|49.26|
> |ASL|48.20|44.02|51.27|
> |EM|47.79|45.12|54.07|
> |PiSA|57.58|51.46|62.43|
> ## I.3&III.1 Relation to MNAR / PUe
> To **I.3**: Our point is not that feature-dependent missingness itself is new. Rather, SalSPML has a different bias structure from PUe-style methods. In PUe, the bias is instance-level: a true positive instance is sampled depends on its features.
> In contrast, SalSPML is single-positive multi-label learning: each instance may have multiple true positive labels, but only one as supervision. **The bias therefore lies in which positive label is retained within a multi-label set, rather than whether a positive instance is observed.**
>
> To **III.1**: Our focus is biased annotations in SPML, not instance-level selection bias in binary classification.
> Our prototype-guided idea may extend to binary settings with similar selection bias, which is beyond the scope of this paper.
> ## II.2 Fixed $k$ is Robust
> To validate the setting of fixed top-k design is stable and reliable, we carry out an experiment on **an adaptive similarity-threshold variant**.
> In this variant, we compute the mean similarity $\mu$ and standard deviation $\sigma$ between each image and all class prototypes, and mask the loss for a class when its similarity exceeds $\mu + w\cdot\sigma$.
> **The results show that** this variant performs comparably to fixed $k=3$, but does not outperform it. Moreover, the adaptive rule rejects mostly 2 labels per sample, which is highly consistent with fixed $k$.
> More details are provided in **Reviewer 9JVr-W2**.
>
> Moreover, in multi-label learning, negative labels are abundant. Rejecting a few extra negatives usually has limited effect, while keeping false negatives in supervision is much more harmful.
> ## II.3 Warm-up and Confirmation Bias
> A two-stage training scheme is common in prior SPML methods, and we set 1 warm-up epoch to limit overfitting.
> A sensitivity study (detailed in **Reviewer 9JVr-W1**) proves warm-up=1 is stable: it outperforms 0 epoch, while longer warm-up will not be better and may even hurt.
> **Moreover, PiSA does not keep warm-up prototypes fixed**: they only provide an initial anchor and are updated online.
> Thus, the results do not support the severe self-reinforcing confirmation-bias scenario.
> ## III.2 Reliability of the Real Dataset
> The real saliency annotations are **only used to construct the weakly supervised training set**. Validation and test sets remain the original **fully labeled benchmarks**, so saliency noise does not affect evaluation.
>
> We also conducted an agreement study on 200 sampled images from COCO and VOC, each re-labeled by an additional annotator. **Using Cohen's Kappa, we obtain 0.76 on COCO and 0.61 on VOC, indicating reliable annotation quality.**
> ## III.3 Failure Cases
> We agree that the paper would be strengthened by adding explicit failure-case analysis, and we will include it in future works.

---

> > ### Author Rebuttal · Reviewer_nE8V · 2026-04-03
> >
> > The additional experiments on hard-sample performance, prototype-to-sample distances, and inter-annotator agreement partially alleviate my concerns on distribution shift (I.2/II.1), Theorem 3.3, and dataset reliability. However, three points remain insufficiently engaged with: (1) The theoretical question in I.1 about how Theorem 3.2's bounds change under stochastic annotation was not addressed; (2) The MNAR/PU positioning (I.3) was dismissed as "beyond scope" without comparative discussion; (3) Failure case analysis was deferred to "future works." These could have been discussed within the rebuttal itself. The rebuttal has improved my view of the paper, but not sufficiently to turn the score to above borderline.

---

> > > ### Author Response · Authors · 2026-04-05
> > >
> > > Thank you for your detailed feedback. We are happy that your concerns on distribution shift (I.2/II.1), **Theorem 3.3**, and dataset reliability have been addressed.
> > >
> > > Due to the limited space in the rebuttal, we were unable to fully elaborate on all of your concerns in the previous rebuttal round. Below, we provide a more detailed response to each point:
> > >
> > > ## (1) **Theorem 3.2** under Stochastic Annotation (I.1).
> > > Introducing stochasticity into **Eq. (3)**, which captures the core bias of salient annotations, **preserves our conclusion.**
> > >
> > > Let $p_{sal}(j|x, Y)$ be the probability that the $j$-th label in sample $x$ with ground-truth $Y$ is annotated as a salient label.
> > > Relaxing the deterministic function into a stochastic annotation distribution, the stochastic population risk $R_{sal}^{stoch}(f)$ in **Eq. (5)** of our manuscript becomes:
> > > $$R_{sal}^{stoch}(f) = E_{(x,Y)}[ \sum_{j \in Y} p_{sal}(j |x, Y) \cdot L(f(x), j) ].$$
> > > Consequently, the annotative bias $\Delta_{bias}(f)$ in **Eq. (6)** transitions to a stochastic form of $\Delta_{bias}^{stoch}(f)$:
> > > $$\Delta_{bias}^{stoch}(f) = E_{(x, Y)}[ \sum_{j \in Y} (1 - p_{sal}(j | x, Y)) \cdot L(f(x), j) ].$$
> > > Even under stochastic cases, the probability of annotating a non-salient object $p_{sal}(j_{non-salient}|x, Y)$, where $j_{non-salient}$ represents a non-salient object, remains systematically and strictly low across annotators. Therefore, $(1 - p_{sal}(j_{non-salient}|x,Y))$ remains strictly positive. **This proves $\Delta_{bias}^{stoch}$ remains bounded away from zero.**
> > >
> > > We will explicitly include this stochastic generalization of **Theorem 3.2** in the revised appendix.
> > > ## (2) Relation to MNAR / PU (I.3).
> > > **It is notable that "beyond scope" refers to the direct adaptation of our PiSA method to binary PU learning tasks falls outside our multi-label focus.**
> > >
> > > We stand by our position that **SalSPML is fundamentally different from MNAR/PU, rendering existing MNAR/PU solutions unsuitable for SalSPML.**
> > >
> > > **Theoretically, SalSPML and MNAR/PU model different probability distributions.**
> > > - MNAR/PU: MNAR/PU methods (e.g., PUe [1]) estimate a propensity score $p(s_c=1|y_c=1, x)$. This formulation assumes that observing a label depends only on the instance features $x$ of that specific category.
> > > - SalSPML: Salient annotation is a saliency-driven mapping from the latent label set $Y$. The annotation distribution is $p(s_c=1 | y_c=1, Y, x)$, as the probability of annotating $s_c=1$ is conditioned on not only the features $x$, but also the entire ground-truth label set $Y$.
> > >
> > > In summary, SalSPML estimates $p(s_c=1 | y_c=1, Y, x)$, taking unobserved positive labels in $Y$ that act as high-dimensional latent variables into account. However, MNAR/PU methods focus on $p(s_c=1|y_c=1, x)$, which overlook the latent joint dependency of $x$ and $Y$.
> > >
> > > **Experimentally, we evaluate MNAR/PU transferability by decomposing the $M$-class task into $M$ independent subproblems.** Specifically, for each class $c$, observed labels are designated as positive (P), while all unobserved labels are treated as unlabeled (U).
> > >
> > > As the code for PUe [1] is not publicly available, we reproduce several baselines reported in the literature [1], including Dist-PU [2], nnPU [3], and nnPUSB [4]. The results are presented in **Table R2**.
> > >
> > > **Table R2.** Comparison of PU-based and baseline methods on real SalSPML datasets (COCO and PASCAL) in terms of mAP. Best results are highlighted in bold.
> > >
> > > | |COCO|PASCAL|
> > > |-|-|-|
> > > |Dist-PU|4.88|80.34|
> > > |nnPU|4.82|49.06|
> > > |nnPUSB|19.02|77.95|
> > > |BCE(Baseline)|47.82|82.46|
> > > |PiSA(Ours)|**59.58**|**87.83**|
> > >
> > > The performance collapse on COCO (80 classes) confirms that independent PU modeling cannot scale to high-dimensional $Y$ where $Y$ severely impacted the modeling of the positive sample sampling distribution.
> > >
> > > ## (3) Failure Case Analysis (III.3).
> > > We provide additional CAM visualizations of failure cases at https://anonymous.4open.science/r/BadCase-B3AD.
> > > Based on qualitative inspection, we categorize the failure cases into two main types, organized in folders `1` and `2`:
> > > 1. Model attends to non-salient objects, but assigns them low confidence scores (below 0.5).
> > > 2. Target objects are severely occluded or lack category features, making attention difficult.
> > >
> > > We will include a detailed discussion of these failure cases in the final version.
> > >
> > >
> > > [1] Wang X, Chen H, Guo T, et al. Pue: Biased positive-unlabeled learning enhancement by causal inference. NeurIPS, 2023.
> > >
> > > [2] Zhao Y, Xu Q, Jiang Y, et al. Dist-pu: Positive-unlabeled learning from a label distribution perspective. CVPR, 2022.
> > >
> > > [3] Kiryo R, Niu G, Du Plessis M C, et al. Positive-unlabeled learning with non-negative risk estimator. NeurIPS, 2017.
> > >
> > > [4]  Kato M, Teshima T, Honda J. Learning from positive and unlabeled data with a selection bias. ICLR, 2019.

---

### Official Review · Reviewer_dcc2 · 2026-03-12

**Soundness:** 3
**Presentation:** 3
**Significance:** 2
**Originality:** 2
**Overall Recommendation:** 4
**Confidence:** 3

**Summary:**

This paper addresses a practical issue in Single-Positive Multi-Label Learning (SPML). While existing SPML methods assume the observed positive label is sampled uniformly at random, the authors argue that human annotators typically provide the most visually salient label, leaving less prominent (but equally valid) labels unobserved. The authors formalize this as Salient SPML (SalSPML) and theoretically prove that this non-random selection introduces an irreducible annotation bias. To mitigate this, they propose PiSA, a method that leverages the high-quality features of salient samples to build reliable class-wise prototypes. These prototypes are then used to guide embedding learning and reject false-negative supervision via similarity measurement. The method achieves state-of-the-art results on newly collected real-world SalSPML datasets and synthetic benchmarks.

**Compliance With Llm Reviewing Policy:**

Affirmed.

**Final Justification:**

It addresses some of my concerns. But I am okay if the paper is rejected.

**Key Questions For Authors:**

Please see the weaknesses in the above section.

**Limitations:**

I expect the authors to address the major concerns regarding the depth of their analysis and the specificity of their proposed method.

**Strengths And Weaknesses:**

Strength:

1. Practical Problem Setting: The formalization of the Salient Single-Positive Multi-Label Learning (SalSPML) setting addresses a highly realistic and practically meaningful annotation bias commonly found in real-world data collection.

2. Theoretical Foundation: The authors provide solid theoretical analyses to quantify the irreducible annotation bias introduced by salient supervision.

Weakness:
1. The Ambiguous Nature of "Salient Bias" and Lack of Decoupled Analysis. The fundamental premise of the paper is that the SalSPML setting introduces a severe performance degradation compared to Random SPML. However, the authors fail to disentangle the exact mechanism driving this degradation. Saliency could affect model performance through three distinct channels, and the current methodology and analysis fall short of addressing them adequately:

(1) Global Class Imbalance vs. Saliency: In natural datasets, size-based or confidence-based saliency inevitably exacerbates global class imbalance (e.g., "person" is inherently larger and more frequently salient than "cup"). The paper does not provide an ablation to verify if the performance drop is genuinely due to the conditional nature of saliency, or simply because SalSPML creates a severe long-tail distribution. If all classes were equally likely to be salient globally, would SalSPML still be harder than Random SPML?

(2) Conditional Suppression (Co-occurrence Bias): Saliency is relative. When highly salient class A co-occurs with class B, B is systematically suppressed and treated as a negative. However, the mathematical formulation of $\Delta_{bias}$ in Eq. (6) treats the missed hard labels ($Y^{hard}$) merely as an additive expectation , largely ignoring the joint distribution $P(Y^{hard} | Y^{salient})$. If conditional suppression is the main culprit, wouldn't a simpler approach like co-occurrence prior reweighting suffice?

(3) The "Feature Domain Shift" Paradox: The authors correctly identify that non-salient instances suffer from visual degradation (occlusion, small relative size). However, the core mechanism, PiSA, constructs class prototypes strictly from the salient embeddings and relies on them to rescue non-salient features via similarity matching . Furthermore, Theorem 3.3 assumes the learned prototype $P_c$ closely approximates the true class mean $\mu_c$ ($||P_c - \mu_c|| \le \epsilon$) . Estimating the true mean of a highly variant distribution using only a heavily biased subset (salient features) is questionable. How can a prototype strictly anchored to "large, unoccluded objects" reliably match "small, heavily occluded objects"?

2. Mismatch Between the Specific Problem and a Generic Solution. The motivation focuses heavily on the relative nature of saliency (an object is missed because a more salient object suppressed it) . However, PiSA primarily acts as a generic Partial Label Learning (PLL) or Positive-Unlabeled (PU) method. It relies on standard techniques (class-wise embeddings , prototype learning , and similarity thresholding ) to find "similar" missing labels. It does not appear to explicitly model the conditional suppression or saliency ranking mechanism during training. How does PiSA differentiate itself from simply applying a strong generic PLL method with a better embedding space?

3. The Hard Heuristic in Prototype-Guided Loss Rejection (PGLR). Eq. (13) uses a top-$k$ selection strategy for loss rejection . The ablation study shows experiments with $k \in \{1, 2, 3\}$ . However, the number of non-salient objects in an image is dynamic. A fixed $k$ across the dataset seems like a rigid heuristic. If $k=3$ is chosen but an image only has 1 missing positive, the method risks falsely rejecting 2 true negative labels. Could the authors discuss the robustness of this heuristic and whether an adaptive, similarity-score-based thresholding would be more appropriate?

---

> ### Author Rebuttal · Authors · 2026-03-31
>
> We thank the reviewer for their careful reading and constructive feedback.
> ## W1. Additional Analysis on the Sources of Difficulty in SalSPML
> The reviewer points out that the difficulties of SalSPML may stem from multiple factors.
> However, we do not attribute all performance degradation of SalSPML to a single bias term, but rather to challenges beyond standard imbalance noise caused by significance. Therefore, we will clarify these factors point by point.
>
> **(1) Global Class Imbalance vs. Saliency.**
> To isolate the impact of class imbalance, we constructed an **Imbalance-SPML** dataset with the same class distribution as SalSPML, but without saliency-guided label selection.
>
> **Table R1.** Comparison between Imbalance-SPML and SalSPML (mAP).
> |||COCO|||Pascal||
> |-|-|-|-|-|-|-|
> ||BCE|ASL|EM|BCE|ASL|EM|
> |Imbalance-SPML|58.68|58.46|59.49|84.31|86.22|87.84|
> |SalSPML|47.82|50.36|50.66|82.46|84.60|86.04|
>
> As shown in **Table R1**, all baselines perform consistently better on Imbalance-SPML than on SalSPML, indicating that identifying unsalient labels is a distinct challenge.
>
> **(2) Conditional Suppression.**
> Conditional suppression is already reflected in our formulas:
> $y^{sal}=\phi(x,Y)$ and $Y^{hard}=Y\setminus{y^{sal}}.$
> Thus, salient annotation depends on $(x,Y)$.
>
> We test the reviewer's suggested baseline: **class-level co-occurrence prior reweighting** using a static prior $P(B\mid A)$. It brings only marginal gains (COCO / Pascal: 47.82 / 82.46 $\rightarrow$ 48.28 / 83.11), still far below PiSA (59.58 / 87.83). This is because because $P(B \mid A)$ is a global statistic prior, whereas label saliency must be judged together with a specific sample. In contrast, PiSA only identifies unsalient labels if the semantically aligned features exist, making it outperforms the suggested baseline.
>
> **(3) The "Feature Domain Shift" Paradox.**
> Occlusion and small object size can cause visual shifts between salient and hard positives. **However, such shifts do not necessarily destroy class-level semantic consistency.**
> **In the proposed model**, PiSA does not require perfect appearance alignment; it only requires hard positives to remain closer to their class prototype than to negatives.
> **In the experiments**, this condition still holds: **the cosine distance between $P_c$ and $\mu_c$ remains small (0.009), and PiSA achieves an 11.99 mAP improvement on hard samples.**
> **Appendix E.3** further visualizes the t-SNE distribution: although salient and hard positives exhibit some shift, they still cluster around the prototype overall and remain clearly separated from negatives, further proving our idea.
> Detailed experiments, including additional prototype-space analysis and hard-example evaluation, are provided in our response to **Reviewer nE8V's WI.2&II.1**.
>
> **The analysis above can be summarized as**:
>
> **(1)** Long-tail imbalance can only explain part of the degradation.
>
> **(2)** Conditional suppression is important but can't be solved by simple co-occurrence priors.
>
> **(3)** Prototypes learned from salient instances still have sufficient semantic transferability to recover difficult positive examples, which is precisely where PiSA excels.
> ## W2. Why PiSA is Well Matched to SalSPML
> **We must clarify that PiSA is not a general PLL/PU module.**
> SalSPML is a single-positive multi-label learning problem with a special supervised structure: the observed label is a salient positive example, which is more reliable and representative, while unsalient objects are harder to identify, and PiSA is designed to handle such unsalient objects.
> It relies on semantic similarity with a reliable prototype, so as to extract visually weak unsalient objects to reduce the unreliability of the prediction.
>
> **Fig. 4** verifies that **in the same embedding space**, replacing the prediction-based method with PiSA's prototype similarity strategy still brings significant performance improvements (mAP improvements of +1.20/+3.50/+4.27 on the VOC dataset; and +0.79/+0.97/+2.09 on the COCO dataset).
>
> ## W3. Empirical Validation of PGLR
> First, the reviewer has a concern that fixed threshold may reject some additional true negatives. However, in multi-label learning, negative labels are abundant. **Rejecting a few extra negatives usually has limited effect, while keeping false negatives in supervision is much more harmful.**
>
> Additionally, to validate the setting of fixed top-k design is stable and reliable, we further carry out an experiment on **an adaptive similarity-threshold variant**.
> In this variant, we compute the mean similarity $\mu$ and standard deviation $\sigma$ between each image and all class prototypes, and mask the loss for a class when its similarity exceeds $\mu + w\cdot \sigma$.
> **The results show that** this variant performs comparably to fixed $k=3$, but does not outperform it. Moreover, the adaptive rule rejects mostly 2 labels per sample, which is highly consistent with fixed $k$.
> More details are provided in **Reviewer 9JVr-W2**.

---

> > ### Author Rebuttal · Reviewer_dcc2 · 2026-04-05
> >
> > I thank the author for the effort in the rebuttal. It addresses some of my concerns. But I am not fully convinced by the answer to W2. The proposed method does not explicitly incorporate designs tailored for saliency, which creates a gap between the proposed setting and solution.

---

> > > ### Author Response · Authors · 2026-04-06
> > >
> > > We sincerely thank the reviewer for the detailed feedback and for raising the score.
> > >
> > > We would like to clarify why explicitly modeling conditional suppression or saliency ranking mechanism is infeasible in SalSPML, and how PiSA’s components are tailored to mitigate the consequences of saliency suppression, differentiating it from generic PLL/PU methods.
> > >
> > > ## Why Not Explicitly Model the Conditional Suppression or Saliency Ranking Mechanism
> > > In real-world scenarios, annotators naturally tend to label the most visually salient objects, which introduces an inherent annotation bias. Intuitively, it might seem appropriate to explicitly model the saliency-based annotation probability $P(s_c=1|x, Y)$, where $s_c$ is a binary indicator of whether category $c$ is an observed salient label, $x$ denotes the input image features, and $Y$ represents the complete ground-truth label set.
> > >
> > > However, under the extremely constrained setting of SalSPML, such explicit modeling faces significant hurdles.
> > > **Fundamentally**, since only a single salient label is observed for each instance, the complete ground-truth set $Y$ remains **a high-dimensional latent variable devoid of direct supervision**.
> > > **What makes matters worse**, we completely lack any objective supervision for the saliency degree itself, such as ground-truth saliency rankings or scores.
> > >
> > > Consequently, given this dual lack of supervision, precisely modeling the conditional suppression or saliency ranking mechanism is highly challenging, as such unconstrained estimation would easily trigger severe confirmation bias and overfitting.
> > >
> > > ## Why PiSA is Specifically Tailored to Bridge the Gap for SalSPML
> > > **Rather than performing estimation under a lack of supervision, PiSA is specifically designed to mitigate the direct consequences of saliency bias through two explicit strategies:**
> > > - **From Suppressed Logits to Semantic Embeddings.** Saliency bias naturally suppresses the logits of non-salient targets. As a result, generic probability-based pseudo-labeling fails. **PiSA addresses this by recovering missing labels in the semantic feature space.**
> > > Even when non-salient targets are visually weak, their deep representations still align with their respective class prototypes. As shown in **Fig. 4**, our similarity-based strategy significantly outperforms generic probability-based methods under SalSPML.
> > > - **Exploiting Saliency Bias for Reliable Anchors.** Saliency bias makes non-salient targets harder to learn, **but it also provides a highly valuable prior**: the observed salient labels are exceptionally clean and representative.
> > > PiSA explicitly leverages this prior. We construct high-quality prototypes exclusively from these reliable salient annotations. These prototypes then act as strong semantic anchors to safely guide the feature learning of the weaker, non-salient targets.
> > >
> > >  **To empirically validate PiSA's capability to mitigate saliency bias, we quantitatively compared the saliency bias reduction of PiSA against generic baselines across multiple datasets in Fig. 5.** The results demonstrate that PiSA maintains a significant advantage over generic baselines, experimentally demonstrating that our method explicitly suppresses the saliency bias.
> > >
> > > We will ensure that the reviewer's suggestions and our proposed revisions are integrated into the final version of the manuscript.

---

### Decision · Program_Chairs · 2026-04-30

**Decision:**

Accept (regular)

**Comment:**

After rebuttal and discussion steps, all reviewers reached a consensus that this paper should be accepted. The work is well-motivated, the proposed method is sound, and both theoretical analysis and extensive experiments convincingly support the claims. Minor issues such as prototype initialization, threshold selection, and scalability should be further refined in the final version.